# LAKEQA: An Exploratory QA Benchmark over a Million-Scale Data Lake

**Haonan Wang** [*] ♣ **Jiaxiang Liu** [*] ♣ **Yurong Liu** ♠ **Austin Senna Wijaya** ♣ **Tianle Zhou** ♣ **Eden Wu** ♠
**Yijia Chen** ♣ **Wanting You** ◇ **Reya Vir** ♣ **Daniela Pinto** ♠ **Grace Fan** ♠ **Yusen Zhang** ♣ **Juliana Freire** ♠
**Eugene Wu** ♣

[*] Equal contribution; order decided by a coin flip.

♣ 👑DAPLab Columbia University    ♠ VIDA New York University    ◇ Barnard College

{hw2983, jl6235, asw2215, yc4719, rrv2116, mz2998, yz5296, ew2493}@columbia.edu,
{yurong.liu, yfw215, daniela.pinto, gf2467, juliana.freire}@nyu.edu, wy2470@barnard.edu

## Abstract

Recent large language models (LLMs) have shown rapid progress in reading-based question answering (QA), where evidence is explicitly provided or can be trivially retrieved. In contrast, real-world questions are often not paired with accurate evidence documents. The useful evidence resides in a massive collection of data lakes, necessitating searching as a prerequisite for answering. However, there is a lack of a comprehensive benchmark that requires searching and reasoning over a large collection of data lakes. To this end, we introduce LAKEQA, a comprehensive benchmark for search-centric question answering over data lakes that jointly emphasizes *searching* and *reasoning* capabilities. LAKEQA is built on a heterogeneous collection of ∼9.5 TB text resources from Wikipedia and open-source government data, spanning structured and unstructured data. To ensure the quality of LAKEQA's tasks, each sample is annotated by at least one Ph.D level expert. Each task requires long-horizon multi-hop reasoning with implicit intermediate steps: agents need to discover the correct document(s) and then compose evidence across sources to produce the answer. Experiment results on seven frontier LLMs have demonstrated that LAKEQA is challenging. For instance, GPT-5.2 achieves only an exact-matching score of 18.37% on LAKEQA. Overall, LAKEQA provides a realistic testbed for developing LLM agents that can both *find* and *analyze* data in modern data lakes.

. Correspondence to: Jiaxiang Liu <jl6235@columbia.edu>, Haonan Wang <hw2983@columbia.edu>.

*Proceedings of the 43rd International Conference on Machine Learning*, Seoul, South Korea. PMLR 306, 2026. Copyright 2026 by the author(s).

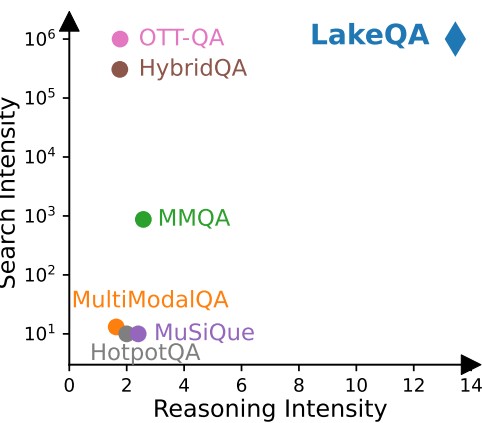

*Figure 1.* The landscape of the search and reasoning intensity in existing QA datasets, measured by the number of documents and reasoning steps, respectively. LAKEQA involves both high search and reasoning intensity.

## 1. Introduction

Recent large language models (LLMs) have shown rapid progress on Question Answering (QA) tasks, including reading comprehension (Rajpurkar et al., 2018; Kwiatkowski et al., 2019; Khashabi et al., 2018; Dua et al., 2019), open-domain question answering (Joshi et al., 2017; Karpukhin et al., 2020), multihop reasoning (Yang et al., 2018; Talmor & Berant, 2018; Trivedi et al., 2022; Wu et al., 2025; Chen et al., 2020b), and tabular question answering (Pasupat & Liang, 2015; Nan et al., 2022; Herzig et al., 2021). These advances have enabled LLMs to effectively serve user needs across diverse QA settings, from answering factual questions over web corpora to reasoning over knowledge bases and curated document collections.

However, these approaches typically assume that the evidence needed to answer a question is either provided upfront or retrievable from a narrow, curated corpus. In contrast, answering questions over heterogeneous data lakes requires the agent to first discover which files are relevant before it can reason over them. Moreover, modern data lakes can

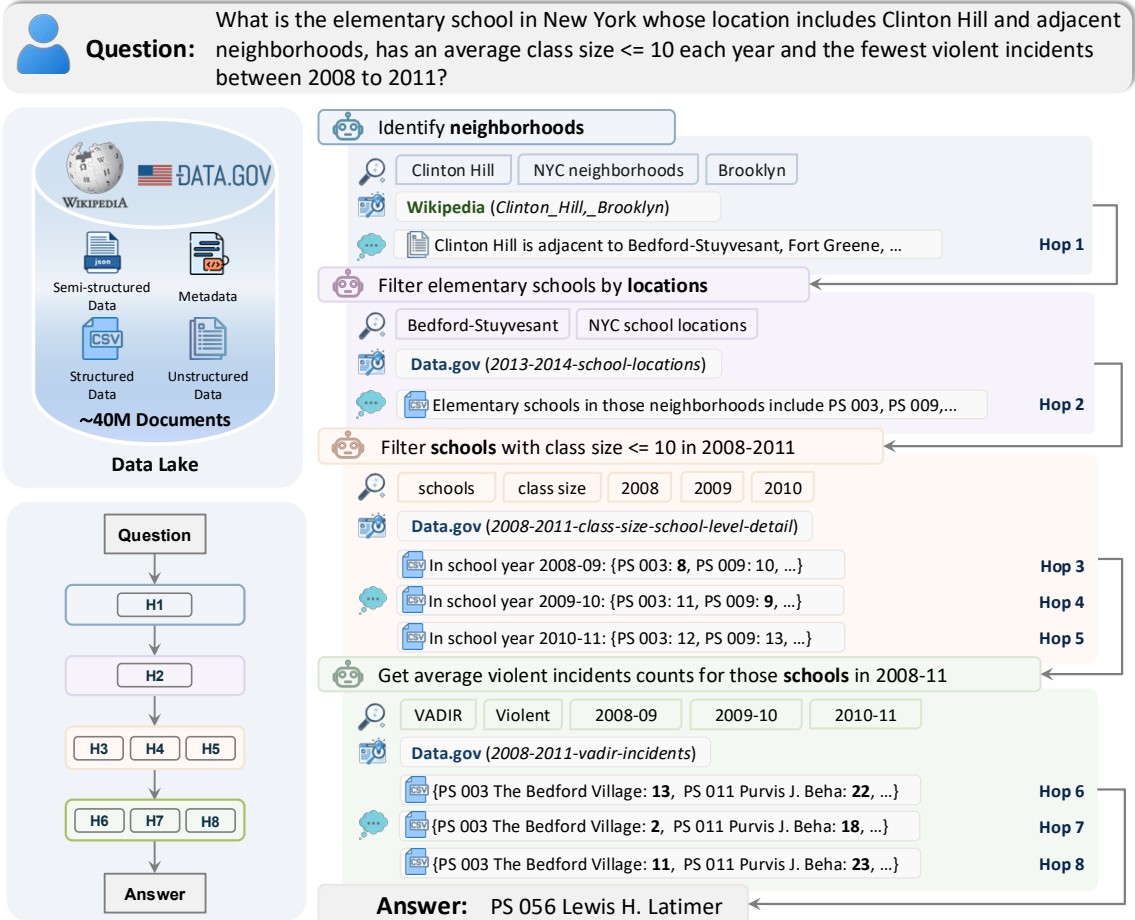

*Figure 2.* An illustrative task in LAKEQA demonstrating multi-hop exploratory question answering over a heterogeneous data lake. Given a complex natural language user question, an agent must iteratively decompose it into a sequence of sub-tasks that each depends on the answers from previous sub-tasks.

contain millions of documents spanning both structured and unstructured formats (Halevy et al., 2016; Chapman et al., 2020). Since evidence for a single question may be scattered across many files, the relevant sources are not known in advance. Effective question answering over data lakes therefore requires iterative search and reasoning over a large search space. This problem has attracted growing interest from both academia (Deng et al., 2024; Khatiwada et al., 2023; Fan et al., 2023b; Zhu et al., 2019; Chapman et al., 2020; Fan et al., 2023a; Nargesian et al., 2019) and industry (OpenAI, 2025; Uber Engineering, 2025; Databricks, 2024; Microsoft, 2024).

To address this real-world task, we formalize **Exploratory Question Answering** (EQA), where an LLM-powered agent starts with a natural-language question and must *repeatedly* reason about missing evidence and search through a *vast* data lake to locate it. Figure 2 shows an example of an EQA task. Starting from the user question, the agent first uses Wikipedia to identify Clinton Hill and its adjacent neighborhoods (Hop 1). It then queries school-location data to enumerate candidate elementary schools in those

neighborhoods (Hop 2), consults class-size records to retain schools whose average class size is at most 10 between 2008 to 2011 (Hop 3-5), and finally uses VADIR incident data to compare the remaining candidates and select the school with the fewest violent incidents (Hop 6-8). Crucially, each hop produces intermediate entities and constraints needed to formulate the next search, so the full retrieval path is not known in advance and errors can propagate across hops.

Challenging EQA tasks should have high *reasoning intensity* and high *search intensity*, two axes first informally introduced in (Java et al., 2025): agents must execute long reasoning chains where later steps depend on the intermediate results of earlier steps, while repeatedly exploring the data lake to discover additional evidence-bearing documents needed across those steps. We follow existing literature (Trivedi et al., 2022) and use the number of documents for a task ($K$), or number of hops, as reasoning intensity and define the number of documents in the data lake ($N$) for search intensity. In the example task in Figure 2, $K = 8$ hops and $N \approx 40M$. While real-world EQA demands large $K$ and $N$ simultaneously, existing benchmarks (Yang et al.,

2018; Talmor & Berant, 2018; Geva et al., 2021; Chen et al., 2020b;a; Trivedi et al., 2022) do not scale both. For example, OTT-QA has a large $N$ but exhibits small $K$, with fewer than 2 documents on average, while MMQA (Wu et al., 2025) increases $K$ but the search space only contains hundreds of documents.

To address this limitation, we propose LAKEQA, a benchmark for EQA that jointly scales the two intensities. For large $N$, LAKEQA is built over a *vast data lake* of ∼9.5 TB spanning ∼40 million heterogeneous documents, including both structured (`csv`, `json`) and unstructured files (`txt`, `pdf`, `html`), with individual tables containing millions of rows and spanning gigabytes. For large $K$, each task in LAKEQA is crafted by human annotators to ensure it consists of multiple dependent reasoning hops grounded in evidence from different documents. Together, as shown in Figure 1, LAKEQA is the first benchmark to combine multi-step reasoning with search at this scale.

We conduct extensive experiments with seven state-of-the-art LLMs spanning both proprietary and open-source model families. Each model is equipped with a tool suite that mirrors realistic data analysis workflows over a data lake. Despite access to search and inspection tools, all models achieve $< 35\%$ end-to-end accuracy over LAKEQA-full, indicating that EQA remains challenging even for frontier LLMs. By analyzing agent traces, we found that failures are dominated by *missing evidence*: models frequently fail to discover or open the gold documents needed to answer the question. Moreover, accuracy degrades sharply as reasoning intensity increases, consistent with long-horizon exploration amplifying error accumulation. Overall, these results indicate that effective search and exploration are the primary bottlenecks for current LLM agents on EQA.

## 2. Related Work

**Multi-Hop QA over Text.** Early benchmarks established compositional reasoning paradigms over text corpora, with limited retrieval scope. COMPLEXWEBQUESTION (Talmor & Berant, 2018) decomposes questions into symbolic operations over knowledge bases (KB) and remained limited to KB-anchored entities. Further, it provides gold supporting paragraphs and does not require search. HotpotQA (Yang et al., 2018) advances multi-hop reasoning over Wikipedia by requiring hyperlinks across multiple documents, and MuSiQue (Trivedi et al., 2022) increased reasoning intensity but provides gold supporting paragraphs and does not require search. STRATEGYQA (Geva et al., 2021) shifts toward *implicit multi-hop reasoning*, in which intermediate steps must be discovered, thereby increasing reasoning intensity within Wikipedia corpora. In contrast, LAKEQA significantly scales both reasoning and retrieval complexity.

**Question Answering over Heterogeneous Data.** Recent benchmarks extend QA beyond text to heterogeneous modalities. FETAQA (Nan et al., 2022) and MMQA (Wu et al., 2025) focus on intra-table or multi-table relational reasoning for tabular QA, but do not require large-scale retrieval. HYBRIDQA (Chen et al., 2020b) combines Wikipedia tables with linked passages for table–text reasoning while providing all relevant context. OTT-QA (Chen et al., 2020a) decontextualizes HYBRIDQA's questions and introduces open-domain retrieval over a large corpus of passages and table segments, while retaining relatively shallow reasoning depth. MultiModalQA (Talmor et al., 2021) also incorporates images, but structures questions with pre-associated context, thus reducing search intensity. Unlike these benchmarks that provide relevant context or operate over curated collections, LAKEQA requires agents to discover evidence from a ∼40 million file collection spanning structured and unstructured data across more diverse sources and domains, as shown in Figure 4.

**Open-Domain Data Discovery.** Recent benchmarks have emphasized search intensity for the problem of multi-hop question answering. BROWSECOMP (Wei et al., 2025) moves beyond static corpora to the open web, thus requiring iterative retrieval. MM-BROWSECOMP (Li et al., 2025) extends BROWSECOMP to multimodal open-domain reasoning across text and images. While these works focus on search in an open web environment, LAKEQA targets data discovery at a data lake scale, where agents must reason over both structured and unstructured data.

The data management community has conducted extensive work on document search over data lakes, focusing on retrieving relevant documents for keywords or tables (Fan et al., 2023a; Chapman et al., 2020; Zhang & Balog, 2020; Brickley et al., 2019; Castelo et al., 2021). Benchmarks for document retrieval over government open data for keyword searches (Kato et al., 2021; Chen et al., 2020c; Zhang et al., 2025a) address the scale and heterogeneity of data lakes (Nargesian et al., 2019). However, they do not consider multi-hop question-answering as a downstream task. In contrast, LAKEQA treats data retrieval as a necessary component of multi-hop question answering, combining high search intensity with high reasoning intensity.

## 3. Benchmark Design

LAKEQA evaluates the capability to discover relevant documents in a large data lake and reason over them across multiple dependent steps. In this section, we formalize the task and describe the construction of LAKEQA.

*Table 1.* Comparison of QA benchmarks. LAKEQA uniquely combines high search intensity, high reasoning intensity, and a realistic large-scale data lake. Text refers to unstructured text. Reas. = Reasoning, Srch. = Search, Ints. = Intensity.

| Benchmark | Modalities | Reas. Ints. | Srch. Ints. | Search Space |
|---|---|---|---|---|
| ComplexWebQuestion (Talmor & Berant, 2018) | Text | 2.90 | ✗ | – |
| HotpotQA (Yang et al., 2018) | Text | 2.00 | ✓ | 10 paragraphs |
| MuSiQue (Trivedi et al., 2022) | Text | 2.40 | ✓ | 10 paragraphs |
| StrategyQA (Geva et al., 2021) | Text | 2.94 | ✗ | Wikipedia corpus |
| BrowseComp (Wei et al., 2025) | Text | – | – | Open Web |
| FeTaQA (Nan et al., 2022) | Table | 3.58 | ✗ | – |
| MMQA (Wu et al., 2025) | Table | 2.58 | ✓ | Curated table collection |
| HybridQA (Chen et al., 2020b) | Text + Table | 1.76 | ✓✓ | 293K paragraphs + 13K Wikipedia tables |
| OTT-QA (Chen et al., 2020a) | Text + Table | 1.77 | ✓✓✓ | 5M paragraphs + 400K Wikipedia tables |
| MultiModalQA (Talmor et al., 2021) | Text + Table + Image | 1.63 | ✓ | ≤ 15 documents |
| MM-BrowseComp (Li et al., 2025) | Text + Image | – | – | Open Web |
| **LAKEQA (our work)** | **Text + Table** | **13.12** | **✓✓✓** | **∼40 million documents** |

## 3.1. Task Formulation and Evaluation

LAKEQA defines a shared environment consisting of a data lake $\mathcal{D} = \{D_1, \ldots, D_n\}$ where each element $D_i$ is an individual document from Data.gov or Wikipedia. Because an arbitrary data lake lacks a unified query language and global schema, LAKEQA provides a fixed set of basic interactive tools $\mathcal{T}$ for discovery, retrieval, and local analysis (Table 8). In each instance of LAKEQA, an agent receives a natural-language question $\mathcal{Q}$ and is required to produce a natural-language answer $\mathcal{A}_{\text{pred}}$.

To solve an instance of LAKEQA, an LLM-based agents must identify a set of relevant documents $\{D_1, \ldots, D_k\} \subseteq \mathcal{D}$, extract supporting facts $\mathcal{F}_i$ from each retrieved document $D_i$, and an agent parametrized by $\theta$ predicts a final answer

$$\mathcal{A}_{\text{pred}} = f_\theta(\{\mathcal{F}_i\}_{i=1}^k, \mathcal{Q})$$

In LAKEQA, we annotated the reference gold answer $\mathcal{A}^*$ and gold relevant documents $\mathcal{D}^* \subseteq \mathcal{D}$. We evaluate end-to-end answer accuracy using Exact Match (EM) based on string matching as well as an LLM-as-a-judge metric, and use $\mathcal{D}^*$ to evaluate evidence grounding.

## 3.2. Data Lake Construction

**Dataset Collection.** Real-world data lakes contain messy, heterogeneous data of various types and sizes—ranging from kilobyte text files to gigabyte-scale tables with millions of rows. To reflect this heterogeneity, we construct our data lake from two complementary open-data sources: Wikipedia and Data.gov. Wikipedia serves as a canonical text corpus for question answering, while Data.gov introduces the diverse and weakly standardized artifacts that characterize real-world public data lakes. For **Wikipedia**, we parse the latest English Wikipedia dump[1] (24.05 GB compressed), extracting each page's wikitext content and associated metadata as a separate document. For **data.gov**,

we build on Harvard LIL's data.gov archive[2] (17.9 TB), a pre-crawled snapshot of catalog records from data.gov. For each catalog record, we harvest all associated artifacts: structured metadata, tabular resources (`csv`, `json`), provider webpages (`html`), and linked reports (`pdf`, `txt`). The storage layout is described in Section D.

**Quality Control.** The Harvard LIL data.gov archive may include duplicate representations of the same resource (e.g., both `csv` and `json`). We deduplicate and retain a single canonical copy per resource, yielding ≈ 40M unique files (≈ 9.5 TB). Details are provided in Section D.

## 3.3. Task Annotation Process

Inspired by the COMPOSITION operation in COMPLEXWEBQUESTIONS (Talmor & Berant, 2018), which treats an entity mention in the question as a variable and replaces it with a sub-question whose answer fills the slot, we adopt subquestion composition as the core primitive for constructing LAKEQA instances. Unlike previous benchmarks built on Wikipedia (Yang et al., 2018; Trivedi et al., 2022), our benchmark draws from a diverse data lake, which does not offer a consistent hyperlink structure or a structured query language for automatic composition. We therefore employ human annotators to create long-horizon multi-hop questions spanning multiple documents through manual COMPOSITION. The full pipeline, illustrated in Figure 3, consists of two stages: *question construction* and *question rewrite*. In the question construction stage, an annotator first selects a document, derives a fact, and reformulates it as a subquestion-answer pair. The annotator then iteratively extends the reasoning chain by finding another document and writing a new subquestion whose required conditions are instantiated by the preceding answers, yielding a subquestion chain with arbitrary hop count ($K$). In the question rewrite stage, the annotator rewrites the subquestion chain into a single natural-language question, replacing document-

---

[1] https://dumps.wikimedia.org/enwiki/latest/enwiki-latest-pages-articles.xml.bz2

[2] https://source.coop/harvard-lil/gov-data

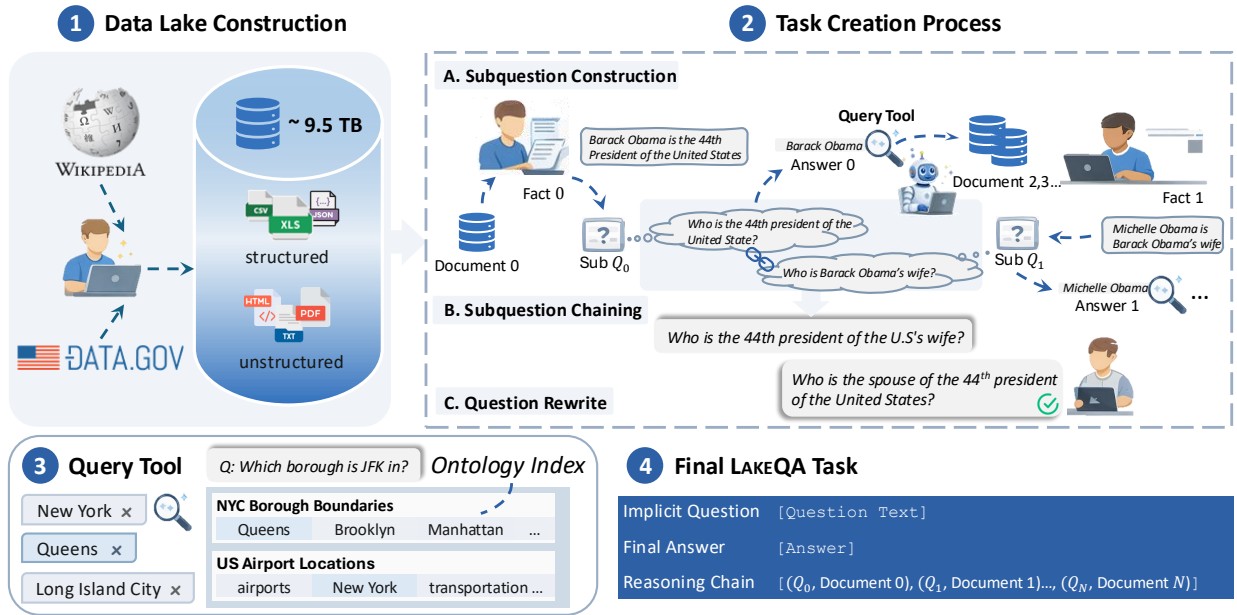

*Figure 3.* Overview of LAKEQA creation process. (1) *Data Lake Construction*: We construct a heterogeneous collection of ∼9.5 TB from Wikipedia and data.gov, including both structured and unstructured data. (2) *Task Creation*: Annotators create multi-hop QA tasks through an iterative annotation process: we first derive facts from a document and reformulate them into subquestions (A), then chain subquestions by finding new facts whose conditions depend on previous answers (B), and finally rewrite the composed question to avoid leaking document identifiers (C). (3) *Query Tool*: Throughout annotation, an ontology-indexed search interface helps annotators efficiently search the large collection. (4) *Final Task Format*: Each task in LAKEQA comprises an implicit question (with reasoning steps hidden), a ground-truth answer, and the reasoning chain documenting the evidence trail.

identifying terms with natural-language descriptions and paraphrasing for fluency. At each stage, independent reviewers verify the output before proceeding.

**Question Construction.** The principle of question construction is that the answer to each subquestion must serve as a prerequisite for the next subquestion. Accordingly, annotators construct a sequence of document-specific subquestion-answer pairs $\{(\mathcal{Q}_i, \mathcal{A}_i)\}_{i=1}^K$ such that, for each $i < K$, $\mathcal{A}_i$ is required to instantiate $\mathcal{Q}_{i+1}$, and further derive $\mathcal{A}_{i+1}$. To begin, an annotator selects an arbitrary document and derives a document-specific subquestion-answer pair $(\mathcal{Q}_1, \mathcal{A}_1)$. For example, starting from the Wikipedia page for the Brooklyn neighborhood, Clinton Hill, the annotator derives the subquestion-answer pair $(\mathcal{Q}_1, \mathcal{A}_1)$ = ("What neighborhoods border Clinton Hill?", {"Navy Yard", "Bedford-Stuyvesant", "Fort Greene", "Prospect Heights", "Williamsburg"}). Then, the annotator searches for another document to construct a new pair $(\mathcal{Q}_2, \mathcal{A}_2)$ such that $\mathcal{Q}_2$ is parameterized by $\mathcal{A}_1$, i.e., the answer to the preceding subquestion provides a necessary condition for deriving the next answer. In our running example, $\mathcal{Q}_2$ asks for the elementary schools located in the neighborhoods in $\mathcal{A}_1$. Iterating this process $K$ times yields a subquestion chain $\{(\mathcal{Q}_i, \mathcal{A}_i)\}_{i=1}^K$, in which each step depends on the answer of the previous step.

**Question Rewrite.** Given the subquestion chain

$\{(\mathcal{Q}_i, \mathcal{A}_i)\}_{i=1}^K$ produced from *Question Construction*, annotators then rewrite each composed question to check two criteria: (1) *naturalness* — the question should read as one a domain expert would naturally pose; and (2) *no leakage* — the question should not expose document-specific identifiers that could trivialize data discovery. Specifically, if the composed question contains source-identifying terms, annotators substitute them with natural-language descriptions derived from the document's metadata, then phrase the result to restore fluency while preserving the original semantics. For example, a reference to `vadir-incidents` would be rewritten as "violent incidents in New York City public schools as "vadir" (Violent and Disruptive Incident Reporting) is in the name of the document.

**Quality Control.** To guarantee the correctness of each task as well as the questions' naturalness, we recruit nine annotators: five computer science Ph.D. students with at least two years of research experience in data management, and four senior computer science undergraduates who passed a qualification screening on data science proficiency. Each task in LAKEQA is reviewed by at least four independent annotators, including at least one Ph.D. annotator.

Quality control follows the same two-stage workflow as benchmark construction. During *Question Construction*, two undergraduates independently verify that each

*Table 2.* LAKEQA statistics.

| Statistics | LAKEQA-full | LAKEQA-mini |
|---|---|---|
| Number of tasks | 1007 | 135 |
| Avg. Wikipedia docs/task | 2.06 | 2.40 |
| Avg. Data.gov docs/task | 11.06 | 12.27 |
| Question Length (max) | 255 | 220 |
| Question Length (avg) | 101.06 | 93.04 |

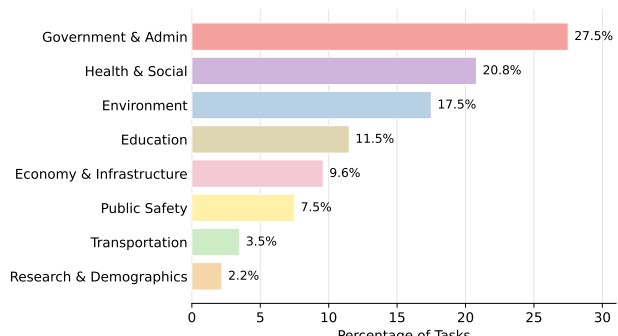

*Figure 4.* Distribution of task domains.

subquestion-answer pair $(\mathcal{Q}_i, \mathcal{A}_i)$ is faithfully derived from its source document, and that for each $i < K$, the answers $\{\mathcal{A}_i\}_{i \leq j}$ correctly serve as a prerequisite for instantiating the next subquestion $\mathcal{Q}_{j+1}$. This ensures that the resulting sequence $\{(\mathcal{Q}_i, \mathcal{A}_i)\}_{i=1}^K$ forms a valid reasoning chain. During *Question Rewrite*, a third undergraduate verifies that the composed question preserves the semantics of the subquestion chain while satisfying the two rewrite criteria: *naturalness* and *no information leakage*. Finally, a Ph.D. annotator adjudicates the previous checks, ensures that the rewritten question is unambiguous, and confirms that the final answer is uniquely determined by the referenced evidence. Tasks that fail any check are revised with corrected evidence trails or removed from the benchmark. Detailed annotation guidelines can be found in Section H.

### 3.4. Benchmark Statistics

We summarize key statistics of LAKEQA in Table 2. We divide our LAKEQA benchmark into two parts, LAKEQA-mini (135 tasks) and LAKEQA-full (1007 tasks), where LAKEQA-mini is a stratified subset of LAKEQA-full that preserves the distribution of reasoning intensity. We measure question length in tokens by tokenizing with the `cl100k-base` encoder (OpenAI, 2022). We derive each task's domain labels from the topic metadata provided by the Data.gov sources for each task. Because a task may span multiple domains, we treat domain assignment as multi-label and count each task toward every associated domain. This yields the domain distribution shown in Section E.

## 4. Experiment

We design experiments based on the LAKEQA dataset to answer the following questions: **RQ1**: How do LLMs perform on EQA tasks? **RQ2**: What is the main reason for LLMs' failure? **RQ3**: How does LLMs' performance change regarding the search and reasoning intensity?

### 4.1. Experimental Settings

**Datasets.** We report our experiments on both LAKEQA-mini and LAKEQA-full. Due to the complexity of LAKEQA, LAKEQA-mini is intended to enable rapid iteration and quick evaluation on EQA, or for those with limited computing resources.

**Models.** We evaluate our benchmark on seven frontier LLMs spanning both proprietary and open-source model families. For proprietary models, we include OpenAI `gpt-5.2` and `gpt-5-mini`, and Anthropic `Claude Haiku 4.5`, `Claude Sonnet 4.5`, and `Claude Opus 4.5`. For open-source models, we include `DeepSeek-R1` and `Llama-3.3-70B-Instruct`. We only evaluate `Claude Opus 4.5` on LAKEQA-mini due to its high cost.

**RAG retrieval baselines.** We additionally evaluate retrieval-augmented variants over document subsets sampled from the data lake, with $N \in \{10k, 25k, 50k\}$ documents. For each document, we sample at most 10K words. We chunk the sampled content using an 800-word window with a 100-word overlap. Each chunk is embedded with `Qwen3-Embedding-0.6B` (Zhang et al., 2025b) and inserted into LanceDB (Pace et al., 2025). For keyword search, we tokenize the sampled text with the `en_stem` tokenizer and build a full-text search index using BM25 (Robertson & Zaragoza, 2009); for vector search, we build an ANN index using `IVF_HNSW` (Malkov & Yashunin, 2020).

We consider two retrieval conditions on top of the baseline discovery and execution tools. In the BM25 condition, an LLM agent is given an additional search tool with signature `(query, k)` where `query` is the keyword search query and `k` specifies the number of top-ranked retrieval results to return. In the HYBRID SEARCH condition, the agent is instead given a search tool with the same signature, which combines BM25 keyword search with HNSW vector search and reranks results using reciprocal rank fusion (RRF), with weight 0.7 assigned to vector search.

**Evaluation Metrics.** We report end-to-end task performance using *Exact Match* (EM) against the ground-truth answer provided and verified by multiple annotators, together with the *wall-clock runtime* and *dollar cost* per task (computed from provider pricing and the model's measured token usage / API billing units). To study the role of search in the evaluation, we additionally evaluate *data discovery*

*Table 3.* Results on the LAKEQA-full.

| Evaluation Methods | End-to-end | | | Accessed Set $\mathcal{D}_{acc}$ | | | Retrieval Set $\mathcal{D}_{ret}$ | | |
|---|---|---|---|---|---|---|---|---|---|
| **Metrics** | **EM(%)↑** | **Runtime(s)↓** | **Cost($)↓** | **P(%)↑** | **R(%)↑** | **F1(%)↑** | **P(%)↑** | **R(%)↑** | **F1(%)↑** |
| *Open Source LLMs* | | | | | | | | | |
| `Llama-3.3-70B` | 5.06 | 86.71 | 0.05 | 0.81 | 0.22 | 0.33 | 0.32 | 4.96 | 0.50 |
| `DeepSeek-R1` | 15.99 | **72.80** | 0.07 | 1.57 | 0.42 | 0.58 | 0.60 | 2.82 | 0.82 |
| *Proprietary LLMs* | | | | | | | | | |
| `Claude-haiku-4.5` | 11.42 | 128.56 | 0.48 | 11.22 | 5.93 | 7.36 | 0.60 | 26.59 | 1.15 |
| `Claude-sonnet-4.5` | **32.87** | 159.02 | 1.42 | **23.39** | **13.04** | **15.69** | **1.41** | **42.36** | **2.64** |
| `GPT-5-mini` | 5.16 | 347.44 | **0.01** | 4.60 | 2.13 | 2.46 | 0.59 | 8.61 | 0.84 |
| `GPT-5.2` | 18.37 | 271.14 | 0.32 | 7.54 | 5.98 | 5.52 | 1.32 | 38.88 | 2.46 |

*Table 4.* Results on the LAKEQA-mini.

| Evaluation Methods | End-to-end | | | Accessed Set $\mathcal{D}_{acc}$ | | | Retrieval Set $\mathcal{D}_{ret}$ | | |
|---|---|---|---|---|---|---|---|---|---|
| **Metrics** | **EM(%)↑** | **Runtime(s)↓** | **Cost($)↓** | **P(%)↑** | **R(%)↑** | **F1(%)↑** | **P(%)↑** | **R(%)↑** | **F1(%)↑** |
| *Open Source LLMs* | | | | | | | | | |
| `Llama-3.3-70B` | 1.48 | 73.57 | 0.04 | 0.00 | 0.00 | 0.00 | 0.22 | 5.39 | 0.40 |
| `DeepSeek-R1` | 6.67 | **72.34** | 0.07 | 1.60 | 0.21 | 0.36 | 0.78 | 4.39 | 1.16 |
| *Proprietary LLMs* | | | | | | | | | |
| `Claude-haiku-4.5` | 11.11 | 128.03 | 0.49 | 12.83 | 6.21 | 8.02 | 0.86 | 29.70 | 1.63 |
| `Claude-opus-4.5` | **45.93** | 141.66 | 2.51 | **35.42** | **20.03** | **24.63** | **2.61** | **54.11** | **4.81** |
| `Claude-sonnet-4.5` | 33.33 | 159.14 | 1.40 | 22.26 | 12.67 | 15.42 | 2.02 | 48.65 | 3.70 |
| `GPT-5-mini` | 2.22 | 388.00 | **0.01** | 5.46 | 1.46 | 2.15 | 0.28 | 6.20 | 0.52 |
| `GPT-5.2` | 17.04 | 264.25 | 0.32 | 7.59 | 4.48 | 4.82 | 2.04 | 44.98 | 3.67 |

*Table 5.* Comparison of evaluation methods on LAKEQA-mini.

| Model | EM | LLM-as-a-Judge | Human |
|---|---|---|---|
| `Claude-sonnet-4.5` | 45/135 | 45/135 | 45/135 |
| `GPT-5.2` | 23/135 | 24/135 | 24/135 |
| `GPT-5-mini` | 3/135 | 3/135 | 3/135 |

*Table 6.* Trajectory-level failure analysis on LAKEQA-mini.

| Failure Type | GPT-5.2 | GPT-5-mini | Claude-sonnet-4.5 |
|---|---|---|---|
| Correct | 24/135 | 3/135 | 45/135 |
| Search Dataset Missing | 31/135 | 112/135 | 20/135 |
| Retrieved Not Selected | 56/135 | 10/135 | 19/135 |
| Retrieved Not Analyzed | 4/135 | 0/135 | 2/135 |
| Wrong after Analysis | 20/135 | 10/135 | 49/135 |

*Table 7.* RAG-style retrieval baselines on LAKEQA-mini document subsets. All values are EM (%).

| Model | Search | 10K | 25K | 50K |
|---|---|---|---|---|
| `Claude-sonnet-4.5` | BM25 | 37.04 | 31.11 | 35.56 |
| `Claude-sonnet-4.5` | Hybrid | 32.59 | 34.07 | 33.33 |
| `Claude-haiku-4.5` | BM25 | 19.26 | 17.78 | 19.26 |
| `Claude-haiku-4.5` | Hybrid | 24.44 | 15.56 | 14.81 |

metric; 62/135 (45.9%) answers are numeric, and 33/135 (24.4%) are temporal, together covering 95/135 (70.4%) of the tasks. Many questions also specify explicit output formats, such as rounding to the nearest integer or reporting dates in MM/DD format.

**Overall Flow.** For each task in LAKEQA, an agent is presented with a natural language question and a set of tools described in Table 8 in Section C. The agent iteratively selects exactly one tool call each round, observes the tool result, and updates its internal state using the full tool-call history. The loop terminates when the agent submits an answer, reaches a turn limit, or times out. We record the final answer reported by an agent for evaluation. Further details, including agent prompts and tool signatures, are provided in Section G.

### 4.2. Results & Analysis

In this section, we discuss and analyze our experiment results reported in Tables 3 and 4.

via document-level precision, recall, and F1 under two document collections induced by the agent's traces: first, the *retrieval set* $\mathcal{D}_{\mathrm{ret}}$ contains the union of all documents returned by the search tools, regardless of whether it was subsequently used by the agent. Second, the *accessed set* $\mathcal{D}_{\mathrm{acc}}$ contains document-IDs the agent actually *opened/queried* towards constructing its answer. Then, we compute precision and recall by comparing $\mathcal{D}_{\mathrm{ret}}$ and $\mathcal{D}_{\mathrm{acc}}$ to the annotated golden set $\mathcal{D}^\star$ for each task. The gap between $\mathcal{D}_{\mathrm{ret}}$ and $\mathcal{D}_{\mathrm{acc}}$ can be interpreted as *reasoning failure*, indicating that the agent mistakenly discards documents relevant to the final question. By contrast, low recall on $\mathcal{D}_{ret}$ can be interpreted as a *search failure*, indicating that the agent was unable to find the relevant documents. We further validate EM with LLM-as-a-judge and human evaluation on LAKEQA-mini. Since many answers are normalized numeric or temporal values, EM is a conservative but generally reliable primary

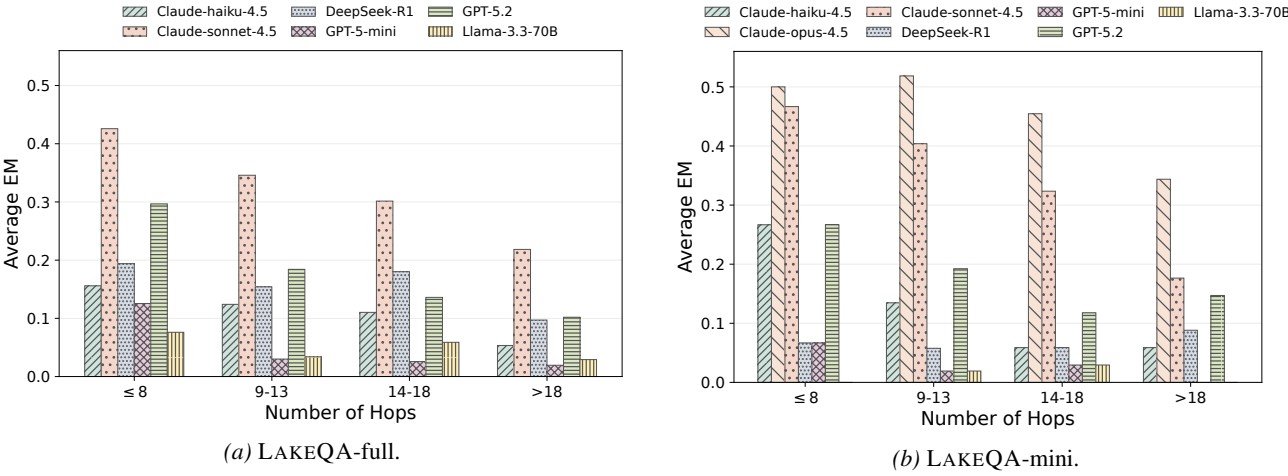

*(a)* LᴀᴋᴇQA-full.  *(b)* LᴀᴋᴇQA-mini.

*Figure 5.* Average EM on LᴀᴋᴇQA stratified by number of hops per task; each bar reports the mean EM over tasks in the corresponding range.

**RQ1: How do LLMs perform on EQA tasks.** Overall, all models achieve relatively low EM, indicating that EQA over a large, heterogeneous data lake remains challenging even for frontier LLMs. On LᴀᴋᴇQA-full, the best end-to-end performance is achieved by Anthropic `Claude-sonnet-4.5` (32.87% EM), followed by `GPT-5.2` (18.37%) and `DeepSeek-R1` (15.99%). On LᴀᴋᴇQA-mini, the strongest model is `Claude-opus-4.5` (45.93% EM), with `Claude-sonnet-4.5` (33.33%) and `GPT-5.2` (17.04%) next, suggesting that increased capability improves EQA but is still far from solving the problem.

Table 7 reports RAG retrieval baselines over 10K, 25K, and 50K document subsets. The results show that adding retrieval tools can improve EM on reduced collections, but the gains are unstable across search methods and collection sizes. This suggests that retrieval quality matters, but retrieval in EQA is not a single-shot step: after each partial result, the agent must infer what information is still missing, formulate the next query needed to find it, and decide whether the retrieved evidence is sufficient for the final answer. Simple BM25 or hybrid search can help with individual retrieval calls, but it does not solve this iterative query-planning problem. We therefore view these results as initial baselines rather than an exhaustive exploration of the RAG design space.

To study the effect of semantic equivalence of task answers in LᴀᴋᴇQA, Table 5 compares EM with LLM-as-a-judge and human evaluation over LᴀᴋᴇQA-mini. Since most LᴀᴋᴇQA answers are constrained by the question format, EM is usually sufficient, and only a small number of cases require semantic judgment. For example, one EM failure occurred when `GPT-5.2` produced "Pittsburgh, PA", whereas the gold answer was "Pittsburgh, Pennsylvania". We further asked human annotators to review the LLM-as-a-judge

decisions, and all decisions were accepted, suggesting that LLM-as-a-judge is sufficient as a supplementary check for rare surface-form mismatches.

Finally, efficiency differs dramatically across models. For example, `GPT-5-mini` is inexpensive per task (0.01$ on LᴀᴋᴇQA-mini and LᴀᴋᴇQA-full) but has the longest runtime, whereas `GPT-5.2` and `Claude-haiku-4.5` are faster but more costly, and `Claude-opus-4.5` delivers the best accuracy on LᴀᴋᴇQA-mini at substantially higher cost. This is because `GPT-5-mini` often writes inefficient code to query large tables, where larger models tend to avoid.

**Takeaway:** EQA presents a three-way trade-off between accuracy, runtime, and dollar cost for LLMs; however, the current Pareto front for the trade-off remains low.

**RQ2: What is the main reason for LLMs' failure.** *Across all models, the main cause of failure comes from not being able to find the gold documents required to answer the question.* Specifically, in both LᴀᴋᴇQA-full and LᴀᴋᴇQA-mini, the recall of $\mathcal{D}_{acc}$ and $\mathcal{D}_{ret}$ are low, where the recall of $\mathcal{D}_{ret}$ is only slightly higher. This gap suggests the bottleneck occurs early: models often fail to even surface the relevant documents during search.

Moreover, there exists a large gap between $\mathcal{D}_{ret}$ and $\mathcal{D}_{acc}$, suggesting that search results are often available but not effectively explored by the agent. For example, on LᴀᴋᴇQA-full, `Claude-sonnet-4.5` achieves 42.36% recall on $\mathcal{D}_{ret}$ but only 13.04% recall on $\mathcal{D}_{acc}$; similarly, `GPT-5.2` achieves 38.88% recall on $\mathcal{D}_{ret}$ but only 5.98% recall on $\mathcal{D}_{acc}$. This indicates that LLM agents often retrieve relevant candidate documents but fail to analyze, or even inspect them when planning subsequent steps. These results suggest that EQA systems may require stronger document-level pointers or guidance mechanisms to help agents decide

which retrieved documents should be analyzed after search.

To further localize failure modes, Table 6 provides a trajectory-level breakdown on LAKEQA-mini. We categorize each trajectory by the furthest stage it reaches, making the buckets mutually exclusive. *Search Dataset Missing* means that none of the required datasets appears in the retrieved set; *Retrieved but Not Selected* means that at least one required dataset is retrieved but no required file is accessed; *Retrieved but Never Analyzed* means that a required file is downloaded but neither inspected nor used in code; and *Analyzed but Still Wrong* means that a required file is inspected or used in code but the final answer is still wrong. This finer breakdown shows that many failures occur before successful document-level analysis begins.

**Takeaway**: Search and exploration (document discovery) is the primary bottleneck; reasoning over found evidence is secondary.

**RQ3: How does LLMs performance change with regard to the search and reasoning intensity.** Figure 5 shows a clear and consistent trend across all evaluated models: *as the number of documents required per task increases, average EM drops substantially.* When tasks require only a small evidence set (e.g., $\leq 8$ documents), models achieve their highest EM, suggesting that the search component is often manageable over tasks with small or moderate reasoning intensity. That is, when the rounds of exploration in the data lake is small, agents are able to reason over what information they are looking for and know what to search to retrieve them. However, as the reasoning intensity increases where tasks expand to involve more documents (e.g., $\geq 14$ and especially $> 18$), performance downgrade drastically. This regime forces search to be tightly interleaved with multi-step reasoning: the agent must repeatedly retrieve, maintain intermediate hypotheses, and integrate evidence across documents. Each additional document effectively adds another dependent step, amplifying error accumulation from imperfect retrieval, missed constraints, and faulty cross-document aggregation, which makes the overall task harder and drives EM down.

**Takeaway**: EQA performance degrades steeply with higher search and reasoning intensity because long-horizon exploration and multi-document integration magnify compounding errors.

## 5. Conclusion

In conclusion, we introduced LAKEQA, a benchmark built up on a million-scale data lake for Exploratory Question Answering (EQA). LAKEQA is designed to examine LLM agents in realistic settings where evidence is not provided and must be discovered through iterative search and reasoning. By constructing LAKEQA over public government documents with analytical statistics and employing dedicated task creation process with multiple annotators, we aim to reduce contamination and ensure high-quality evaluation. Our experiments with seven frontier LLMs show that EQA remains far from solved: even with access to search and inspection tools, end-to-end accuracy remains low, and performance drops sharply as tasks require longer reasoning chains and more evidence sources. Trace-based analysis further reveals that the main bottleneck is document discovery–models frequently fail to retrieve and open the gold documents–while conservative exploration strategies that yield high precision but low recall are heavily penalized.

## Acknowledgements

This research was partially supported by National Science Foundation grants (NSF 1527765, 1564049, 1845638, 1740305, 2008295, 2106197, 2103794, and 2312991), NSF-OAC 2411221, DARPA ASKEM (HR0011262087) and ARPA-H BDF, an IBM PhD fellowship, and support from Modal, as well as DAPLab corporate support in the form of funding and/or compute from Amazon, IntellectAI, Infosys, Tidalwave, Veris, Shopify, Microsoft, Thinking Machines, Dandy, Perplexity, and Daytona. The views and conclusions presented here are those of the authors and should not be interpreted as representing the official positions of the funding organizations.

## Impact Statement

This paper presents a benchmark designed to advance research in machine learning, particularly in question answering, retrieval, and reasoning over large-scale data lakes. The benchmark is constructed entirely from publicly available data and is intended as an evaluation resource rather than a deployed system. We do not foresee societal or ethical impacts beyond those commonly associated with progress in data-driven machine learning methods.

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

## A. Task description for Annotators to Characterize Searching and Reasoning Intensity

Let $\mathcal{D}$ be the collection of data source, the goal is to evaluate some statistics about tasks in existing QA benchmarks. For each task with a pair of question and answer $(Q, A)$, we define the underlying *question decomposition graphs*, or QDGs, as a collection of directed acyclic graphs (DAGs) $\{\mathcal{G}_i\}$. Each DAG $\mathcal{G}_i = (\mathcal{V}, \mathcal{E})$ semantically represents a minimal reasoning trace of deriving the answer $A$ from the question $Q$. The set of nodes $\mathcal{V}$ encodes questions and the set of directed edges $\mathcal{E}$ encodes dependencies between the questions. Specifically, $\mathcal{V} = \{Q, A\} \cup \{Q_i\}$ where $Q$ is the source node, $A$ is the sink node and $\{Q_i\}$ are sub-questions.

Each sub-question $Q_i$ is an *atomic* question, meaning that $Q_i$ can be answered by *exactly* one piece of data source, in each benchmark, one piece of data source could be a paragraph, a table, or a single image. A directed edge $Q_i \rightarrow Q_j$ exists *if and only if* $Q_j$ depends on the answer of $Q_i$ (guarantees minimality). For example, the question $Q$: "Who is the spouse of the $44^{th}$ president of United States?" can be decomposed into a QDG $\mathcal{G}_1$ as $Q \rightarrow Q_1 \rightarrow Q_2 \rightarrow A$ where $Q_1$: "Who is the $44^{th}$ president of United States?" and $Q_2$: "Who is the spouse of $A_1$?" where $A_1$ is the answer of $Q_1$. Finally, the final answer $A$ only depends on $Q_2$.

In the case where there exists different ways to answer $A$ based on $Q$, for example, the same $Q$ can be decomposed via a single sub-question $Q_3$: "Who is the $44^{th}$ first lady of United States?". This induces a second QDG $\mathcal{G}_2$ as $Q \rightarrow Q_3 \rightarrow A$.

Tasks in existing QA benchmarks are constructed by a QDG, and we would like to quantify the reasoning complexity of the tasks. As the first step, we would like to collect the following statistics

- (S1) The smallest number of nodes across all QDGs.

- (S2) The shortest directed path from $Q$ to $A$ over all QDGs.

- (S3) For each QDG, calculate the maximum-cardinality minimal $Q$-$A$ cut, and return the minimal over all QDGs.

These statistics above are for each task in a benchmark. Over all tasks of the benchmark, calculate the **min, max, mean and median** for each of S1, S2 and S3, and report what are the set of data sources, e.g. a paragraph in Wikipedia, a table in Wikipedia or an image in Wikipedia.

Existing work is either open domain or closed domain, for open domain, each question maps to a search, for close domain, each sub-question maps to a data source. For example, each sub-question in COMPLEXWEBQUESTION maps to an internet search that will return the answer to this sub-question, it is unclear how many data sources each sub-question maps to. On the other hand, each sub-question in HotpotQA maps to a single data source.

## B. Extended Related Work

### B.1. COMPLEXWEBQUESTION (Talmor & Berant, 2018)

[Text, Open Domain] COMPLEXWEBQUESTIONS targets open-domain questions backed by a search engine. The paper highlights a decomposition framework to parse the input natural language question via a computation tree. The leaf nodes of the computation tree consists of questions that are directly answerable from the search engine; internal nodes apply operators—conjunction (set intersection), comparatives (filters like >), superlatives (argmax/argmin) and composition (join)—to combine leaf answers. For example, the leaves "Which films were directed by Christopher Nolan?" and "Which films star Leonardo DiCaprio?" each return a set of films; the parent node applies conjunction to yield the composite question "Which Nolan films star Leonardo DiCaprio?". In collecting tasks for the benchmark, the authors start from WEBQUESTIONSSP questions, where each question is paired with a simple SPARQL query. COMPLEXWEBQUESTIONS generate more complex SPARQL queries by applying one operation in {CONJUNCTION, COMPARATIVE, SUPERLATIVE, COMPOSITION} to a SPARQL query from WEBQUESTIONSSP.

- CONJUNCTION takes two SPARQL queries to ensure their intersection is an non-empty set, so the output SPARQL simply concatenates involving conditions from the involving queries.

- COMPARATIVE intakes a SPARQL query and find an attribute where the denotation of the SPARQL query possess in common, and apply a filter on that attribute.

- SᴜᴘᴇRʟᴀᴛɪᴠᴇ is mostly similar to CᴏᴍᴘᴀRᴀᴛɪᴠᴇ except that instead of applying a filter, we find an argmax/argmin.

- CᴏᴍᴘᴏSɪᴛɪᴏɴ starts with a SPARQL query $r$, finds an entity $e$ in $r$, and replace that entity with another question whose answer is $\{e\}$ by looking at the KB and finding unique identifiers of $e$.

After arriving at the SPARQL for each task, AMT workers are employed to translate them into natural language questions.

### B.2. HotpotQA (Yang et al., 2018)

[Text, Retrieval, Explicit Multi-hop Reasoning] HotpotQA is a reading comprehension benchmark that evaluates question answering capabilities of an agent over information retrieval and multi-hop reasoning. Unlike ComplexWebQuestion, whose questions are dependent on the knowledge graph of Wikipedia, which forces the questions to only involve entities from the knowledge graph, which is an incomplete source. HotpotQA proposes a method based on hyperlinks between Wikipedia pages through the following procedures

- The authors first extract the first paragraphs of each Wikipedia page. Then, they treat those paragraphs as nodes and build a hyperlink graph by adding an edge between two nodes if one paragraph contains a hyperlink to another.

- Bridge entities: annotators are presented with a pair of paragraphs linked by an edge in the hyperlink graph in the first step, and propose questions requiring both paragraphs to answer.

- Comparison: the authors curate 42 lists of similar entities from Wikipedia (each list contains many paragraphs) and present two paragraphs from the same list to annotators to create questions like "Who has played more games in NBA, Kobe Bryant or Micheal Jordan?"

Aside from answering questions where the exact context needed are given, HotpotQA allows testing for harder tasks where distractor paragraphs are included in the question to test for robustness and the full Wikipedia paragraphs (5M paragraphs) are included in the question to test for relevant information retrieval.

### B.3. MᴜSᴇQᴜᴇ (Trivedi et al., 2022)

[Text, Multi-hop Reasoning, Compositional] MᴜSɪQᴜᴇ addresses the "shortcut" problem in HᴏᴛᴘᴏᴛQA, where models often bypass the intended reasoning chain by relying on single-hop clues or entity overlap. To mitigate this, MᴜSɪQᴜᴇ introduces a systematic bottom-up construction process. Instead of starting with a complex question and decomposing it, the authors begin with a large pool of 2-hop, 3-hop, and 4-hop reasoning graphs composed of connected single-hop questions from existing datasets (NᴀᴛᴜRᴀʟ QᴜᴇSᴛɪᴏɴS and HᴏᴛᴘᴏᴛQA).

### B.4. HʏBRɪᴅQA (Chen et al., 2020b)

[Text, Table, Explicit Multi-hop Reasoning] HʏBRɪᴅQA One key contribution of HʏBRɪᴅQA is that it integrates both tables and text into the context of its questions. The data source of HʏBRɪᴅQA consists of webtables as tabular datasets and paragraphs as text data from Wikipedia. To create tasks for HʏBRɪᴅQA, annotators are given HITs (human intelligence task) where each HIT consists of a single webtable alone with paragraphs linked by hyperlinks in the webtable's cells[3]. For each HIT, annotators are tasked to create 6 questions that requires information relying on both tabular and textual information to answer. The questions are created based on the following three atomic reasoning chains:

- Table → Passage chain: first uses table-wise operations (equal/greater/less/first/last/argmax/argmin) to locate a certain tuple in the table, and retrieves a text span from the passage in the hyperlink of that tuple.

- Passage → Table chain: reverse of the first type, first retrieves a paragraph, and asks about a tuple with a hyperlink that points to this paragraph.

- Passage → Table → Passage: same as the Passage → Table chain, but hops back to another Passage (i.e. another hyperlinked cell in the same tuple).

---

[3]HʏBRɪᴅQA crop at most the first 12 sentences from the introduction paragraph of the Wikipedia page, and small webtables (5-20 rows and 3-6 columns) with hyperlinked cells over 35% of its total cells. This ends up with 13000 webtables.

There are three more types of tasks in HYBRIDQA, which corresponds to the CONJUNCTION, COMPARATIVE and SUPERLA-TIVE operators in (Talmor & Berant, 2018) to compose more complex questions based on questions created via the three atomic reasoning chains.

### B.5. OTT-QA (Chen et al., 2020a)

[Text, Table, Retrieval] OTT-QA builds on top of HYBRIDQA by reusing HYBRIDQA's questions while requiring an additional retrieval step to search for relevant tables and text. To do so, OTT-QA *decontectualize* tasks in HYBRIDQA by removing context-dependent keywords in the natural language questions. For example, "the players" in a HYBRIDQA question is context dependent because the table in the context is about "Netherlands players". But this is ambiguous in OTT-QA when relevant context to answer the question requires retrieval. Additionally, the authors of OTT-QA decompose each table into several table segments where each table segment consists of a tuple, the table's headers, metadata, statistics of the original table, i.e. min/max of a column (table documentations). Together, the candidate pool of retrieval consists of 5M passages and 5M table segments.

### B.6. MultiModalQA (Talmor et al., 2021)

[Text, Table, Image, Explicit Multi-hop Reasoning] MultiModalQA extends existing reading comprehension datasets such as Natural Questions (NQ) (Kwiatkowski et al., 2019), BoolQ (Clark et al., 2019), and HotpotQA (Yang et al., 2018) by incorporating multimodal contexts involving texts, tables and images into its questions. To construct the benchmark, annotators first create single-hop, single-modality, and persistent questions—those whose answers are unlikely to change over time. For instance, given an image of the Statue of Liberty, a persistent question might ask what the statue is holding. MultiModalQA adopts text-based questions from existing reading comprehension datasets such as HotpotQA. Finally, more complex tasks are formed by linking single-hop, single-modality questions when their referenced entities coincide. For example, a paragraph stating "Barack Obama was born in Honolulu, United States" and a table listing "Barack Obama as the 44th President of the United States" share the same Wikipedia entity, Barack Obama, and can therefore be linked to form a question requiring information from both the paragraph and the table.

### B.7. STRATEGYQA (Geva et al., 2021)

[Text, Open Domain, Implicit Multi-hop Reasoning, Binary Response] STRATEGYQA addresses a key limitation of prior QA benchmarks, where all information needed to answer a question is explicitly stated in the question. Instead, it evaluates implicit multi-hop reasoning: the intermediate steps are not spelled out and must be discovered through exploration and retrieval. To construct the dataset, annotators begin with a seed concept and a target yes/no answer, then craft a strategy question whose solution requires composing several atomic facts, each independently verifiable in Wikipedia but not explicitly mentioned in the question. To ensure feasibility (and avoid ungrounded decompositions), annotators also specify, for every step, a candidate Wikipedia page where the fact can be supported; the released data includes these implicit facts and linked source paragraphs as optional intermediate supervision. The benchmark thus tests open-domain retrieval, composition, and reasoning over implicit evidence—beyond surface cues or single-pass reading.

### B.8. FETAQA (Nan et al., 2022)

[Table, Free-form Response, Wikipedia] FETAQA extends existing table QA benchmarks whose answers are typically short answers evaluated by exact matching by introducing long, informative free-form answers grounded in a single Wikipedia table. To construct such questions, FETAQA starts from ToTTo (?), a large-scale table-to-text dataset containing naturally written descriptions fully grounded in Wikipedia tables along with supporting cells highlighted. It then filters ToTTo instances to keep tables of moderate size and descriptions whose highlighted cells span more than a single row or column. Given each instance, annotators are tasked to write a question whose answer is a (possibly slightly edited) description derived from optionally modifying the sentence, table contents, or highlighted region to yield natural question–answer interactions. For automatic evaluation of generated answers, FETAQA reports n-gram overlap metrics (sacreBLEU, ROUGE-1/2/L, METEOR) as well as semantic similarity metrics (BERTScore and BLEURT).

### B.9. BROWSECOMP (Wei et al., 2025)

[Open Domain, Retrieval] BROWSECOMP is an open-domain deep research benchmark. It consists of 1,266 extremely challenging "needle-in-a-haystack" questions requiring multi-step web searches to find the answer. In contrast to the difficult

to answer questions, the results are short facts that can be easily verified. The benchmark was constructed by inverting a typical question-creation process: human annotators started with a known fact (the target answer) and then added multiple specific qualifiers or constraints to the query until that fact became the unique solution. To ensure these tasks could not be solved via shortcuts, annotators are tasked to verify that simple search-engine queries (up to 5 attempts) cannot directly reveal the answer and SOTA models (e.g. GPT-4 and OpenAI's deep research agent) failed to solve each question. Further, if another human could find the answer within 10 minutes, the task was revised with additional criteria to increase its difficulty.

### B.10. MM-BROWSECOMP (Li et al., 2025)

[Multimodal, Open Domain, Retrieval] MM-BROWSECOMP extends text-only web browsing benchmarks such as BROWSECOMP to evaluate multimodal web research capabilities. It consists of 224 challenging questions that require agents to retrieve and reason over both textual and visual content encountered during browser-style navigation, where crucial evidence is often embedded in images or videos and cannot be recovered through text-only search. The benchmark follows a similar inverted question-creation process as BROWSECOMP and includes a checklist-based evaluation that verifies whether agents complete essential multimodal reasoning steps, revealing substantial gaps in current multimodal browsing systems.

### B.11. MMQA (Wu et al., 2025)

[Multi-Table, Multi-Hop, Tabular QA] MMQA is a multi-table question answering benchmark designed to evaluate retrieval and multi-hop reasoning over interconnected tables. Unlike mMltiModalQA datasets that integrate text and images, this benchmark focuses on reasoning across multiple relational tables, requiring models to identify relevant tables, understand their structural relationships (e.g., key-based joins), and synthesize evidence across hops. The benchmark evaluates performance on table retrieval, relational reasoning, and downstream QA or text-to-SQL tasks, highlighting limitations of current models in complex tabular reasoning settings.

## C. Tool Interface

*Table 8.* Tool interface available to agents in LAKEQA.

| Tool | Input | Output / Purpose |
|---|---|---|
| search(*query*) | keyword or tag string | Returns a ranked list of document IDs relevant to the query. |
| listdata(*document-ids*) | list of document IDs | Lists the files available under each document ID directory. |
| download(*document-ids*) | list of document IDs | Downloads all files for each document ID into a per-task local sandbox. |
| inspect(*path*) | file path (e.g., <id>/<file>) | Returns the first $k$ characters (a lightweight preview). |
| query(*path*, *q*) | file path and query $q$ | Queries over locally downloaded data (e.g., filter, aggregate). |

## D. Data Collection

### D.1. Background

This section describes the data sources of LAKEQA, we form a data lake by collecting data from open data repositories including Wikipedia and data.gov. We use the latest English Wikipedia dump[4] and Harvard LIL's data.gov archive[5]. Our final goal is to create question-answering tasks (QA-tasks) requiring multi-hop reasoning and multiple data sources (different Wikipedia pages and data sources from data.gov). To do so, we would need to compose information from each data source (i.e. one data source contains information like Law X was modified on date Y and another data source contains information like Study Z was conducted on date Y, thus date Y can be used to connect these two pieces of information together.) An LLM agent will assist on this step with annotators to verify the validity of the created questions.

### D.2. Task

Each webpage from Wikipedia and data.gov is considered a data source. Please (1) download the data from the provided links in the footnote; (2) preprocess each data source, place them under a folder that is representative of the data source (i.e. title of the Wikipedia or data.gov webpage) and upload all folders into the S3 bucket you created in the pre-step; and (3) parallelize the data downloading, preprocessing and uploading pipeline to handle the large data source ($\approx$ 20GB from Wikipedia and $\approx$ 15 TB from data.gov). Here are (partial) instructions on how to preprocess the data sources, let me know if you encounter any other situation you are uncertain about.

- A data source might contain binary files and pictures that does not contain useful information in creating QA-tasks – remove those files under the folder.

- The raw data source contains html files (and potentially files in other formats) with plenty of redundancy – convert them into txt files by removing redundant characters (i.e. html blocks shall be converted into $text$, you can use the `BeautifulSoup` package under the `bs4` library in Python to do so) and extra white spaces (i.e. /n, /t, etc.), etc.

After you are done, tell me the name of the S3 bucket you created. The deadline is this Saturday. Please don't hesitate to ping me if you have any question.

## E. Distribution of Domains

The distribution of the benchmark tasks across data.gov theme categories are reported in Table 9 and a finer granularity category classification is reported in Table 10.

*Table 9.* Distribution of benchmark tasks across data.gov theme categories.

| Theme Category | Tasks |
|---|---|
| Government & Admin | 428 |
| Environment | 272 |
| Transportation | 54 |
| Health & Social | 324 |
| Research & Demographics | 35 |
| Economy & Infrastructure | 149 |
| Public Safety | 116 |
| Education | 179 |

## F. Agent Interface

**Tool Implementations.** All data access tools operate over a fixed S3 data lake bucket (`lakeqa-yc4103-datalake`) with two namespaces: `wikipedia/` and `datagov/`. Credentials are loaded via environment variables, and all downloads

---

[4]https://dumps.wikimedia.org/enwiki/latest/enwiki-latest-pages-articles.xml.bz2
[5]https://source.coop/harvard-lil/gov-data

*Table 10.* Mapping of data.gov dataset themes to 8 main categories. Task count indicates the number of benchmark tasks containing datasets from each category.

| Main Category | Original Themes | Tasks |
|---|---|---|
| Government & Admin | Administration & Finance, Budget, Campaign Finance, Ethics and Elections, City Government, Civic Vitality & Governance, Elections & Politics, Finance, Finance/Tax/Property, Government, Government & Finance, Government Administration, Government and Taxes, Government-Wide Support, Licenses/Permits, Local Government, Politics, Procurements and Contracts, Regulation, Regulatory, Revenue & Expense, Taxes & Tax Credits | 428 |
| Environment | Boundaries, Energy & Environment, Energy and Environment, Environment, Geography, Geospatial, Locations and Maps, Maps, Natural Resources, Utilities and City Services | 272 |
| Transportation | Aviation, Public Transit, Railroads, Roadways and Bridges, Transportation | 54 |
| Health & Social | 500 Cities & Places, Child & Adult Welfare, Community/Recreation, COVID-19, Disability Compensation, Family & Health, Health, Health & Human Services, Health and Human Services, Health Insurance, Heart Disease & Stroke Prevention, Human Services, Medicare, National Service, NCHS, Nutrition, Parks & Recreation, Physical Activity, Recreation, Recreation and Culture, Smoking & Tobacco Use, Social Services, State Drug Utilization, Unwinding, Vaccinations | 324 |
| Research & Demographics | Community Demographics, Demographics, Research and Statistics | 35 |
| Economy & Infrastructure | Agriculture, Built Environment, Business, Business and Economy, City Infrastructure, Economic Development, Economic Statistics, Economy and Demographics, Employment, Housing, Housing & Development, Housing and Development, Housing and Real Estate, Labor, Planning & Zoning, Public Works, Sales & Distribution, Sanitation, Wireline | 149 |
| Public Safety | Courts, Equity & Justice, Public Safety, Public Safety and Preparedness | 116 |
| Education | Education, Enrollment, Quality | 179 |

are stored in a per-session sandbox directory.

- `search(prefix)`: Performs an S3 prefix search in both namespaces using `list_objects_v2` with `Prefix` and `Delimiter='/'`. The tool returns dataset identifiers corresponding to directory names under each namespace.

- `search_keyword(keyword)`: Issues external keyword searches to the Wikipedia API and the data.gov `package_search` endpoint to propose candidates, then validates each candidate against S3 by checking for a dataset prefix. Candidates are ranked by a lightweight token overlap score that combines query coverage and token density, and the top results are returned.

- `list_files(dataset_id)`: Lists objects under `<namespace>/<dataset_id>/` using `list_objects_v2` and returns relative file paths and sizes (bounded by a maximum count).

- `download(dataset_id, file_path)`: Downloads a specific object to the local sandbox via `s3.download_file`. The tool creates the required directory structure and returns the local path and file size.

- `inspect_file(dataset_id, file_path)`: Retrieves object metadata with `head_object` and reads the first 64KB using a ranged `get_object` request. It infers a delimiter from the first line (comma, tab, pipe, or semicolon), extracts header columns, and, for JSON-like lines, attempts to parse top-level keys. The tool returns metadata only (no raw content).

- `execute_code(code)`: Executes Python in the sandbox with working directory set to the sandbox root. Common libraries (`pandas`, `json`, `csv`, `os`, `glob`, `re`, `Path`) are preloaded, and two helper variables are provided: `SANDBOX_DIR` and `FILES`. The tool captures stdout/stderr, returns errors and tracebacks, and enforces a timeout to prevent inefficient code.

- `get_sandbox_info()`: Enumerates all files in the sandbox via a recursive walk and reports paths and byte sizes.

- `submit_answer(answer, reasoning, sources)`: Terminates the episode and records the final answer and cited sources. The framework expects answers in a normalized bracketed format (e.g., `[123]`).

## G. Evaluation Prompt

*Listing 1.* System prompt and tool schemas used by our data analysis agent.

```
DEFAULT_SYSTEM_PROMPT = """"You are a data analysis agent working with PUBLIC GOVERNMENT
    DATASETS (data.gov, census, etc.).

## HOW THIS WORKS - READ CAREFULLY
This is an INTERACTIVE system. You output ONE tool call, then STOP. The system executes
    your tool and returns the REAL result. Then you are called again to pick the next
    tool.

DO NOT:
- Output multiple tool calls
- Simulate or hallucinate results (e.g., {"result": ...} or {"error": ...})
- Continue the conversation yourself
- Make up data

ONLY output a single JSON object with your tool call. The system handles execution.

## DATA ACCESS (each step = one tool call)
- search(prefixes) or search_keyword(keywords) $\to$ returns dataset_ids (identifiers,
    not data)
  - Pass a list of strings: search(["climate", "weather"]) or search_keyword(["police",
      "crime"])
- list_files(dataset_ids) $\to$ returns file paths in datasets
  - Pass a list of strings: list_files(["dataset1", "dataset2"])
- download(files) $\to$ downloads files to sandbox (max 5 per call)
  - Pass a list of {dataset_id, file_path}: download([{"dataset_id": "...", "file_path":
      "..."}])
- execute_code(code) $\to$ runs Python on downloaded files (use print()!)
- submit_answer() $\to$ when ready to submit final answer
```

```
## CRITICAL: VERIFY DATA SOURCES
Dataset names can be misleading! Example: "traffic-incidents-2020" could be from Chicago,
    NYC, or any other city.
ALWAYS check metadata before using a dataset:
- Download and read metadata files (e.g., metadata.json, catalog.txt) to find the actual
    source
- Look for: publisher, source, city/state, geographic coverage, agency name
- Verify the data matches what the question asks for (correct city, agency, time period)
- Two datasets with similar names may cover completely different locations!

## TIPS
- use search_keyword for semantic matching, SINGLE word preferred
- Always print() in execute_code to see output
- Check actual column names and date formats in the data
- Use full dataset for final answer, not just samples
- Answer format: [value] only, no labels or units

## TURN AND TIME LIMITS
- You have LIMITED TURNS. The system will show you remaining turns.
- There is also a TIME LIMIT. Do not waste time on excessive exploration.
- After inspecting files, IMMEDIATELY run execute_code to analyze data.
- If running low on turns or time, prioritize submitting your best answer."""

# Tool schemas for native tool calling (works with both OpenAI and Claude/Bedrock)
TOOL_SCHEMAS = [
    {
        "name": "search",
        "description": "Find datasets by name prefixes. Returns dataset identifiers (not
            data). Search multiple prefixes at once.",
        "parameters": {
            "type": "object",
            "properties": {
                "prefixes": {
                    "type": "array",
                    "items": {"type": "string"},
                    "description": "List of prefixes to search for (e.g., ['austin-police',
                        'climate', 'weather'])"
                }
            },
            "required": ["prefixes"]
        }
    },
    {
        "name": "search_keyword",
        "description": "Semantic keyword search across datasets. Search multiple keywords
            at once. Returns ranked dataset identifiers.",
        "parameters": {
            "type": "object",
            "properties": {
                "keywords": {
                    "type": "array",
                    "items": {"type": "string"},
                    "description": "List of keywords to search for (e.g., ['police', 'crime',
                        'traffic'])"
                },
                "limit": {
                    "type": "integer",
                    "description": "Max results to return (default 20)",
                    "default": 20
                }
            },
            "required": ["keywords"]
        }
    },
```

```
{
    "name": "list_files",
    "description": "List files within datasets. Use dataset_ids from search results.
        List multiple datasets at once.",
    "parameters": {
        "type": "object",
        "properties": {
            "dataset_ids": {
                "type": "array",
                "items": {"type": "string"},
                "description": "List of dataset identifiers from search results (e.g.,
                    ['Barack_Obama', 'climate-data'])"
            }
        },
        "required": ["dataset_ids"]
    }
},
{
    "name": "download",
    "description": "Download files from datasets to the local sandbox for analysis. Max
        5 files per call.",
    "parameters": {
        "type": "object",
        "properties": {
            "files": {
                "type": "array",
                "items": {
                    "type": "object",
                    "properties": {
                        "dataset_id": {"type": "string", "description": "Dataset
                            identifier"},
                        "file_path": {"type": "string", "description": "Path to file within
                            the dataset"}
                    },
                    "required": ["dataset_id", "file_path"]
                },
                "description": "List of files to download (max 5). Each with dataset_id
                    and file_path.",
                "maxItems": 5
            }
        },
        "required": ["files"]
    }
},
{
    "name": "inspect_file",
    "description": "Inspect a file to see its structure, columns, and sample data
        without downloading.",
    "parameters": {
        "type": "object",
        "properties": {
            "dataset_id": {
                "type": "string",
                "description": "Dataset identifier"
            },
            "file_path": {
                "type": "string",
                "description": "Path to file within the dataset"
            }
        },
        "required": ["dataset_id", "file_path"]
    }
},
{
    "name": "execute_code",
```

```
              "description": "Execute Python code to analyze downloaded files. Use print() to see
                  output. pandas, json, pathlib are available. IMPORTANT: Only use files you have
                  successfully downloaded - use the exact local_path returned by the download
                  tool.",
          "parameters": {
              "type": "object",
              "properties": {
                  "code": {
                      "type": "string",
                      "description": "Python code to execute. Use the exact local_path from
                          download results. Do NOT guess file paths."
                  }
              },
              "required": ["code"]
          }
      },
      {
          "name": "submit_answer",
          "description": "Submit your final answer when you have computed the result.",
          "parameters": {
              "type": "object",
              "properties": {
                  "answer": {
                      "type": "string",
                      "description": "The final answer. Format: just the value, e.g., '12345'
                          not '12345 incidents'"
                  },
                  "reasoning": {
                      "type": "string",
                      "description": "Brief explanation of how you computed the answer"
                  }
              },
              "required": ["answer"]
          }
      }
]
```

## H. Annotation Guideline

Each task shall be a question that requires searching over different data sources across Wikipedia/data.gov to answer. In each task.json, each node represents a dataset, and fact is factual information obtained from that dataset, the subquestion is a question constructed by reversing the fact so it be answered via the fact . Finally, the final question shall be answerable by chaining the subquestions together. What you will need to do: Make sure each task involves at least one dataset from data.gov (no all-wikipedia task). If not, mark them and skip the following steps. For each node , verify that the fact and the subquestion are sound and complete, and each fact is grounded in the source file located in S3.. Each fact is of the shape "a set of entities" satisfies "condition", sound means everything in "a set of entities" satisfies "condition", and complete means "a set of entities" are everything satisfies "condition". For example, the fact "Barack Obama" is "the 44th president of USA" is both sound and complete, while the fact "Barack Obama" is "a democratic US president" is sound but not complete, because the subquestion is then who is a democratic US president, whose answer is not deterministic. Pay attention to the wording of the fact, sometimes the fact obtained from a tabular datasets just uses the column name but this could be WRONG, for example 'crime rate' could refer to 'firearm crime rate' or 'violent crime rate' (this information is sometimes in the metadata) but this ambiguity makes the question unanswerable. Pay attention to facts without a time/location qualifier. For those cases, (a) make the language in the question precise and (b) add the metadata file as a node if this information is not self-contained in the tabular dataset. If some nodes are incorrect, mark them and skip the following steps. After verifying each node is correct, the next step is to rewrite the final question . We want questions that looks natural and does not directly hint which data source to search for. The reasoning chain can be viewed as a directed acyclic graph with nodes as vertices (nodes + one for the final question + one for answer ) and edges are dependencies of between vertices, i.e. nodes depend on the answer of subquestion from previous nodes, if a node does not depend on any other nodes, it depends on the final question. For this graph, add three parameters. k the number of hops, which is the longest path from final question to

answer. n is the number of vertices, and d is the largest number of vertices a vertex depends on.

Some FAQs as examples during annotation: Some answers to subquestions might be incomplete, but the important thing is if it affects the reasoning chain. For example, two subquestions 1. who is Democratics US President and 2. who is US president after 2000 , and in a hop, we take an intersection between the two subquestions: who is a Democratics US President after 2000 , as long as the answer to the intersection is correct, the task is fine – because eventually each task is consist of the final question and its answer, all subquestion and facts are just for the sake of creating the final question. An excellent question about question chaining: because all we need is the final question, the references like (from the intersection of nodes 1-4) is a place holder for annotators to easily chain questions together. For example, first node has something like Who is the US president born in Hawaii and the second node has something like who is the first lady of ¡answer of first node¿ (please refer to the reasoning-chain field of each task to better understand the logic flow of each task). A derivative of 2 is the problem of skip nodes: we are creating all those subquestions and facts in order to chain the subqeustions together, meaning that the answer to previous nodes questions becomes conditions in latter nodes. it is important that the first node's answer is necessary to answer the second node's question. A skipping node is something like who is the first lady of ¡Who is the US president born in Hawaii¿ and is African American because in this case first lady + African American becomes deterministic. We should try to avoid obvious skipping nodes.

## I. Example Task Annotation

*Listing 2.* Complete JSON annotation artifact for a 5-hop LᴀᴋᴇQA task.

```
{
 "question": "What was the 2017-18 total student enrollment in the school district of
     the city that hosts the land-grant university in the only Eastern Washington county
     that borders Idaho AND had 2020 census population under 300,000, among counties with
     school districts achieving ELA performance $\geq$68% for All Students, All Grades in
     all four Washington Report Card Assessment releases from 2014-15 through 2017-18?",
 "final_question": "What is the city that hosts the land-grant university in the only
     Eastern Washington county that borders Idaho and had under 300,000 in 2020? It also
     consistently performed well on English Language Arts assessments between 2014 and
     2018 (>=68%). What was the total student enrollment in the school district of that
     city?",
 "answer": "2941",
 "nodes": {
  "1": {
   "source": "datagov/report-card-assessment-data-2014-15-school-year/files/rows.txt",
   "fact": "In the Washington Report Card Assessment Data 2014-15 release, 21 school
       districts had ELA PercentMetStandard >= 68% for All Students, All Grades,
       including districts in King, Clark, Pierce, Whitman, Thurston, and Spokane
       counties.",
   "subquestion": "Which Washington school districts had ELA performance >= 68% for All
       Students, All Grades from 2014-15 to 2017-18? (Filter:
       OrganizationLevel=District; TestSubject=ELA; StudentGroup=All Students;
       GradeLevel=All Grades; PercentMetStandard>=68.)",
   "answer": [
    "Camas School District (Clark)",
    "Lake Washington School District (King)",
    "Dieringer School District (Pierce)",
    "Colfax School District (Whitman)",
    "Mercer Island School District (King)",
    "Griffin School District (Thurston)",
    "Carbonado School District (Pierce)",
    "Freeman School District (Spokane)",
    "Shoreline School District (King)",
    "St. John School District (Whitman)",
    "Issaquah School District (King)",
    "Northshore School District (King)"
   ],
   "sound": false,
   "complete": false,
   "validation_explanation": "The answer should instead be: [ \"Riverside School
       District\", \"Index Elementary School District\", \"Port Townsend School
```

```
        District\", \"Woodland School District\", \"Deer Park School District\", \"Damman
        School District\", \"Camas School District\", \"Lake Washington School
        District\", \"Blaine School District\", \"Chehalis School District\", \"Dieringer
        School District\", \"Colfax School District\", \"Mercer Island School District\",
        \"Griffin School District\", \"Carbonado School District\", \"Freeman School
        District\", \"Starbuck School District\", \"Shoreline School District\", \"St.
        John School District\", \"Issaquah School District\", \"Northshore School
        District\"]",
    "revision_subquestion": "Which Washington school districts had ELA performance >=
        68% for All Students, All Grades from 2014-15?"
  },
  "2": {
    "source": "datagov/report-card-assessment-data-2015-16-school-year/files/rows.txt",
    "fact": "In the Washington Report Card Assessment Data 2015-16 release, 77 school
        districts had ELA PercentMetStandard >= 68% for All Students, All Grades,
        including districts in all 6 counties from the 2014-15 results.",
    "subquestion": "Which Washington school districts had ELA performance >= 68% for All
        Students, All Grades from 2014-15 to 2017-18? (Filter:
        OrganizationLevel=District; TestSubject=ELA; StudentGroup=All Students;
        GradeLevel=All Grades; PercentMetStandard>=68.)",
    "answer": [
      "Camas School District",
      "Lake Washington School District",
      "Dieringer School District",
      "Colfax School District",
      "Mercer Island School District",
      "Griffin School District",
      "Carbonado School District",
      "Freeman School District",
      "Shoreline School District",
      "St. John School District",
      "Issaquah School District",
      "Northshore School District",
      "and 65 others"
    ],
    "sound": false,
    "complete": false,
    "validation_explanation": "Answer should not include \"and 65 others\", expand it
        instead: [\"Edmonds School District\",\"Fife School District\",\"West Valley
        School District (Spokane)\",\"Bainbridge Island School District\",\"Sedro-Woolley
        School District\",\"Walla Walla Public Schools\",\"Enumclaw School
        District\",\"Hockinson School District\",\"Snohomish School District\",\"Mercer
        Island School District\",\"Grandview School District\",\"Lake Washington School
        District\",\"Granite Falls School District\",\"Colfax School
        District\",\"Evergreen School District (Stevens)\",\"Camas School
        District\",\"Oak Harbor School District\",\"Issaquah School District\",\"Bellevue
        School District\",\"Clover Park School District\",\"St. John School
        District\",\"Quillayute Valley School District\",\"Almira School
        District\",\"Shoreline School District\",\"South Kitsap School District\",\"Pasco
        School District\",\"Northshore School District\",\"Riverside School
        District\",\"Snoqualmie Valley School District\",\"Tahoma School
        District\",\"Chehalis School District\",\"Yakima School District\",\"Ephrata
        School District\",\"Dieringer School District\",\"North Mason School
        District\",\"University Place School District\",\"Wilbur School
        District\",\"Spokane School District\",\"Central Kitsap School
        District\",\"Olympia School District\",\"Ritzville School District\",\"Port
        Angeles School District\",\"Clarkston School District\",\"Ridgefield School
        District\",\"Anacortes School District\",\"Blaine School District\",\"Garfield
        School District\",\"Riverview School District\",\"Freeman School
        District\",\"Oakesdale School District\",\"Sumner-Bonney Lake School
        District\",\"Colton School District\",\"Damman School District\",\"Wenatchee
        School District\",\"Mead School District\",\"Peninsula School
        District\",\"Everett School District\",\"East Valley School District
        (Spokane)\",\"Lake Stevens School District\",\"Griffin School District\",\"Vashon
        Island School District\",\"Evaline School District\",\"Steilacoom Hist. School
```

```
              District\",\"Carbonado School District\",\"Port Townsend School
              District\",\"Index Elementary School District\",\"Pullman School
              District\",\"Rosalia School District\",\"Davenport School District\",\"Lynden
              School District\",\"Liberty School District\",\"Stanwood-Camano School
              District\",\"Tumwater School District\",\"Bellingham School District\",\"Selkirk
              School District\",\"Sequim School District\",\"Sunnyside School District\"]",
         "revision_subquestion": "Which Washington school districts had ELA performance >=
              68% for All Students, All Grades from 2015-2016?"
       },
       "3": {
         "source": "datagov/report-card-assessment-data-2016-17-school-year/files/rows.txt",
         "fact": "In the Washington Report Card Assessment Data 2016-17 release, 67 school
              districts had ELA PercentMetStandard >= 68% for All Students, All Grades,
              including all 14 districts that qualified in 2014-15.",
         "subquestion": "Which Washington school districts had ELA performance >= 68% for All
              Students, All Grades from 2014-15 to 2017-18? (Filter:
              OrganizationLevel=District; TestSubject=ELA; StudentGroup=All Students;
              GradeLevel=All Grades; PercentMetStandard>=68.)",
         "answer": [
           "Camas School District",
           "Lake Washington School District",
           "Dieringer School District",
           "Colfax School District",
           "Mercer Island School District",
           "Griffin School District",
           "Carbonado School District",
           "Freeman School District",
           "Shoreline School District",
           "St. John School District",
           "Issaquah School District",
           "Northshore School District",
           "and 55 others"
         ],
         "sound": false,
         "complete": false,
         "validation_explanation": "Answer should not be in 2014-2015. Should also not have
              'and 55 others'. It should instead be: [\"Enumclaw School District\",\"Snohomish
              School District\",\"Centralia School District\",\"Mercer Island School
              District\",\"Prosser School District\",\"East Valley School District
              (Spokane)\",\"Clover Park School District\",\"Clarkston School
              District\",\"Bainbridge Island School District\",\"Colfax School
              District\",\"Lake Washington School District\",\"Sequim School
              District\",\"Chehalis School District\",\"Edmonds School District\",\"Camas
              School District\",\"Oakesdale School District\",\"Wenatchee School
              District\",\"Almira School District\",\"Lakewood School District\",\"Sultan
              School District\",\"West Valley School District (Spokane)\",\"Yakima School
              District\",\"Issaquah School District\",\"Tahoma School District\",\"North River
              School District\",\"Spokane School District\",\"Fife School
              District\",\"Grandview School District\",\"Bellevue School District\",\"Dieringer
              School District\",\"Snoqualmie Valley School District\",\"Oak Harbor School
              District\",\"Peninsula School District\",\"Shoreline School
              District\",\"Northshore School District\",\"University Place School
              District\",\"Bickleton School District\",\"Sumner-Bonney Lake School
              District\",\"Carbonado School District\",\"Olympia School District\",\"Damman
              School District\",\"Central Kitsap School District\",\"South Kitsap School
              District\",\"Pullman School District\",\"Quillayute Valley School
              District\",\"St. John School District\",\"Granite Falls School
              District\",\"Spokane International Academy\",\"Anacortes School District\",\"Lake
              Stevens School District\",\"Rosalia School District\",\"Paterson School
              District\",\"Mead School District\",\"Coupeville School District\",\"Wilbur
              School District\",\"Sunnyside School District\",\"Everett School
              District\",\"Riverview School District\",\"Vashon Island School
              District\",\"Freeman School District\",\"Hoquiam School District\",\"LaCrosse
              School District\",\"Davenport School District\",\"Adna School
              District\",\"Griffin School District\",\"Kennewick School District\",\"Medical
```

```
                  Lake School District\"]",
    "revision_subquestion": "Which Washington school districts had ELA performance >=
        68% for All Students, All Grades from 2016-2017?"
  },
  "4": {
    "source": "datagov/report-card-assessment-data-2017-18-school-year/files/rows.txt",
    "fact": "In the Washington Report Card Assessment Data 2017-18 release, 56 school
        districts had ELA PercentMetStandard >= 68% for All Students, All Grades,
        including all 13 districts from the 2014-15 intersection.",
    "subquestion": "Which Washington school districts had ELA performance >= 68% for All
        Students, All Grades from 2014-15 to 2017-18? (Filter:
        OrganizationLevel=District; TestSubject=ELA; StudentGroup=All Students;
        GradeLevel=All Grades; PercentMetStandard>=68.)",
    "answer": [
      "Camas School District",
      "Lake Washington School District",
      "Dieringer School District",
      "Colfax School District",
      "Mercer Island School District",
      "Griffin School District",
      "Carbonado School District",
      "Freeman School District",
      "Shoreline School District",
      "St. John School District",
      "Issaquah School District",
      "Northshore School District",
      "and 44 others"
    ],
    "sound": false,
    "complete": false,
    "validation_explanation": "Should not include 44 others. The answer should be:
        [\"University Place School District\",\"Enumclaw School District\",\"Mercer
        Island School District\",\"Bainbridge Island School District\",\"Lake Washington
        School District\",\"Camas School District\",\"Sunnyside School
        District\",\"Colfax School District\",\"Issaquah School District\",\"Blaine
        School District\",\"Pullman School District\",\"Snoqualmie Valley School
        District\",\"Bellevue School District\",\"Carbonado School
        District\",\"Sumner-Bonney Lake School District\",\"Northshore School
        District\",\"Dieringer School District\",\"Colton School District\",\"Almira
        School District\",\"Evergreen School District (Stevens)\",\"Sedro-Woolley School
        District\",\"Shoreline School District\",\"Garfield School District\",\"Wilbur
        School District\",\"Tahoma School District\",\"Olympia School District\",\"St.
        John School District\",\"Freeman School District\",\"Snohomish School
        District\",\"Palouse School District\",\"Davenport School District\",\"Central
        Kitsap School District\",\"Oakesdale School District\",\"Peninsula School
        District\",\"Grapeview School District\",\"Riverview School
        District\",\"Coulee-Hartline School District\",\"Vashon Island School
        District\",\"Ridgefield School District\",\"LaCrosse School
        District\",\"Anacortes School District\",\"Steilacoom Hist. School
        District\",\"Everett School District\",\"Coupeville School District\",\"Griffin
        School District\",\"Centralia School District\",\"Spokane School
        District\",\"Mead School District\",\"Edmonds School District\",\"Stanwood-Camano
        School District\",\"Fife School District\",\"Seattle School District No.
        1\",\"Lake Stevens School District\",\"Endicott School District\",\"Tumwater
        School District\",\"Selkirk School District\"]",
    "revision_subquestion": "Which Washington school districts had ELA performance >=
        68% for All Students, All Grades from 2017-2018?"
  },
  "5": {
    "source": "wikipedia/Eastern_Washington/content.txt",
    "fact": "The Eastern Washington article lists Whitman and Spokane among the counties
        in Eastern Washington.",
    "subquestion": "Among counties from <hop 1 answer: King, Clark, Pierce, Whitman,
        Thurston, Spokane>, which are in Eastern Washington?",
    "answer": "Whitman and Spokane",
```

```
      "sound": false,
      "complete": false,
      "validation_explanation": "Data is true and accurate. However, there seems to be a
          missing node from node 4 to node 5. The intersection of all nodes in node 4 are
          still 12 districts. Yet, they somehow all transform into the 6 counties that
          contain those 12 districts any sources.",
      "revision_subquestion": "Add nodes 4 and 5 to convert the districts into counties.
          Here are the districts: Camas School DistrictCarbonado School DistrictColfax
          School DistrictDieringer School DistrictFreeman School DistrictGriffin School
          DistrictIssaquah School DistrictLake Washington School DistrictMercer Island
          School DistrictNorthshore School DistrictShoreline School DistrictSt. John School
          District"
    },
    "6": {
      "source": "wikipedia/Whitman_County,_Washington/content.txt",
      "fact": "According to the Wikipedia article for Whitman County, Washington,
          'Adjacent counties' include 'Benewah County, Idaho - northeast', 'Latah County,
          Idaho - east', and 'Nez Perce County, Idaho - southeast', confirming Whitman
          County borders Idaho. 'As of the 2020 census, the population was 47,973.'",
      "subquestion": "Among counties from <hop 2 answer: Whitman, Spokane>, which county
          borders Idaho AND has 2020 census population under 300,000?",
      "answer": "Whitman County: borders Idaho (YES, per adjacent counties list),
          population 47,973 (YES, under 300,000) $\to$ MEETS BOTH CRITERIA",
      "sound": true,
      "complete": true,
      "validation_explanation": "Population 47,973 (line 1), borders with Idaho line 48-50"
    },
    "7": {
      "source": "wikipedia/Spokane_County,_Washington/content.txt",
      "fact": "According to the Wikipedia article for Spokane County, Washington,
          'Adjacent counties' include 'Bonner County, Idaho - northeast', 'Kootenai County,
          Idaho - east', and 'Benewah County, Idaho - southeast', confirming Spokane County
          borders Idaho. The 2020 census population was 539,339.",
      "subquestion": "Among counties from <hop 2 answer: Whitman, Spokane>, which county
          borders Idaho AND has 2020 census population under 300,000?",
      "answer": "Spokane County: borders Idaho (YES, per adjacent counties list),
          population 539,339 (NO, exceeds 300,000) $\to$ DOES NOT MEET BOTH CRITERIA",
      "sound": true,
      "complete": true,
      "validation_explanation": "population was 539,339 (line 1). Borders idaho (lines
          80-82)"
    },
    "8": {
      "source": "wikipedia/Pullman,_Washington/content.txt",
      "fact": "According to the Wikipedia article for Pullman, Washington, Pullman is the
          most populous city in Whitman County and is home to Washington State University,
          a public research land-grant university.",
      "subquestion": "Which city hosts the land-grant university in <hop 3 answer: Whitman
          County>?",
      "answer": "Pullman hosts the land-grant university (WSU) in Whitman County",
      "sound": true,
      "complete": true,
      "validation_explanation": "Line 3"
    },
    "9": {
      "source": "datagov/report-card-enrollment-2017-18-school-year/files/rows.txt",
      "fact": "According to the Washington Report Card Enrollment Data 2017-18, Pullman
          School District (in Whitman County) had 2,941 total students enrolled for All
          Grades.",
      "subquestion": "What is the 2017-18 total student enrollment for the school district
          in <hop 4 answer: Pullman>? (Filter: OrganizationLevel=District;
          DistrictName=Pullman School District; GradeLevel=All Grades; column=All
          Students.)",
      "answer": "2941",
      "sound": true,
```

```
      "complete": true,
      "validation_explanation": "True and accurate by query."
    }
  },
  "reasoning_chain": [
    "HOP 1 (d=4, WA Report Card Assessment Data 2014-15 through 2017-18 - nodes 1-4):",
    " Node 1: Districts with ELA >= 68% in 2014-15 $\to$ 21 districts in 6 counties",
    " Node 2: Districts with ELA >= 68% in 2015-16 $\to$ 77 districts",
    " Node 3: Districts with ELA >= 68% in 2016-17 $\to$ 67 districts",
    " Node 4: Districts with ELA >= 68% in 2017-18 $\to$ 56 districts",
    " Intersection: 12 districts in 6 counties (King, Clark, Pierce, Whitman, Thurston,
      Spokane)",
    "",
    "HOP 2 (d=1, Wikipedia Eastern Washington - node 5):",
    " Node 5 (Wikipedia Eastern Washington): Whitman and Spokane are Eastern Washington
      counties",
    " Result: Whitman and Spokane are the Eastern Washington counties from hop 1",
    "",
    "HOP 3 (d=2, Wikipedia county pages - nodes 6-7):",
    " Node 6 (Wikipedia Whitman_County): Borders Idaho (YES), pop 47,973 $\to$ MEETS BOTH
      CRITERIA",
    " Node 7 (Wikipedia Spokane_County): Borders Idaho (YES), pop 539,339 $\to$ DOES NOT
      MEET (pop > 300k)",
    " Result: Whitman County (REQUIRES hop 2 to know which counties to check)",
    "",
    "HOP 4 (d=1, Wikipedia Pullman - node 8):",
    " Node 8 (Wikipedia Pullman): Pullman is home to WSU (land-grant) in Whitman County",
    " Result: Pullman hosts the land-grant university (REQUIRES hop 3)",
    "",
    "HOP 5 (d=1, data.gov enrollment - node 9):",
    " Node 9 (WA Report Card Enrollment 2017-18): Pullman School District = 2,941
      students",
    " Result: 2941 (REQUIRES hop 4 to know the city is Pullman)",
    "",
    "Final answer: 2941"
  ],
  "annotated_by": "[annotator]",
  "annotation_timestamp": "[timestamp]"
}
```

