# OpenReview forum: "LakeQA: An Exploratory QA Benchmark over a Million-Scale Data Lake"
_ICML.cc/2026/Conference — ICML 2026 regular_

### Official Review · Reviewer_sVyN · 2026-02-28

**Soundness:** 3
**Presentation:** 3
**Significance:** 3
**Originality:** 3
**Overall Recommendation:** 4
**Confidence:** 3

**Summary:**

The paper introduces LAKEQA, a benchmark designed to evaluate Exploratory Question Answering (EQA), where agents must navigate a massive ~9.5 TB data lake to discover and reason over distributed information. Unlike existing QA datasets that provide evidence or use narrow search spaces, LAKEQA requires agents to iteratively identify missing evidence and search across approximately 40 million heterogeneous documents, including structured and unstructured formats like CSVs, PDFs, and HTML. The benchmark features expert-annotated tasks requiring long-horizon multi-hop reasoning, often necessitating the discovery of a single relevant document among millions. Experiments on seven frontier LLMs highlight the significant challenge LAKEQA poses, with even advanced models like GPT-5.2 achieving an exact matching score of only 14.73%, establishing a realistic testbed for future developments in large-scale data discovery and analysis.

**Compliance With Llm Reviewing Policy:**

Affirmed.

**Final Justification:**

After considering both the paper and the authors’ rebuttal, I maintain my Weak Accept recommendation. The paper offers a solid and practically relevant benchmark contribution, and I believe it will be useful to the community. The rebuttal addressed my main concerns sufficiently, particularly by clarifying the distinctive value of the data-lake setting and strengthening the empirical support for the paper’s claims.

**Key Questions For Authors:**

1. Please check for typographical errors. Specifically, the sentence immediately preceding Section 3.4 Benchmark Statistics appears to be incomplete.

**Limitations:**

yes

**Strengths And Weaknesses:**

**Strengths**

1. The paper is well-structured and easy to understand.
2. The authors address the challenging task of Exploratory Question Answering by introducing LAKEQA, which requires agents to perform QA across a massive dataset of approximately 9.5 TB of tabular and textual data.

**Weaknesses**

1. **Distinction from Existing Benchmarks:** The distinction between LAKEQA and benchmarks like BROWSECOMP remains unclear. While the authors state that LAKEQA targets "data discovery at a data-lake scale" rather than an "open web environment," they fail to provide a compelling rationale for this design choice. The specific advantages or unique challenges of the data-lake scale over the open web are not sufficiently discussed.
2. **Insufficient Analysis of Experimental Results:** The paper lacks a granular discussion of the findings. Specifically:
   - **Agent Interface (Appendix E):** There is no discussion regarding the rationale behind the specific toolset design. Furthermore, data on tool-calling frequency and ablation studies showing how individual tools impact overall performance are missing.
   - **Evaluation Scope:** Following the methodology of BROWSECOMP and MM-BROWSECOMP, the authors should consider testing official, tool-enabled model services available on web platforms. This would evaluate the holistic capability of the agent system rather than limiting the assessment to model performance on a fixed set of tools.

**Note:** If the authors can provide a convincing justification for the data-lake scale design (Weakness Point 1), I would be open to increasing my score.

---

> ### Author Rebuttal · Authors · 2026-03-31
>
> We thank the reviewer for the constructive suggestions. We address each point below.
>
> **1. Distinction from open-web benchmarks such as BrowseComp.**
> We will make this distinction clearer in the revision. Our goal is not to replace open-web benchmarks, but to study a different problem: **enterprise-style exploratory QA**, where the challenge is to find and combine evidence scattered across a large heterogeneous data lake rather than browse the public web.
>
> LakeQA differs from open-web benchmarks such as BrowseComp in two main ways. **(1)** Unlike BrowseComp, which mainly emphasizes locating and extracting self-contained facts, LakeQA is more **analytic** than extractive: many tasks require data transformation, aggregation, filtering, ranking, comparison, and set intersection over tables. As a result, the agent must reason not only about where evidence is located, but also about **which statistic** the question is asking for and **how it should be computed**, often with the help of metadata. **(2)** This leads to a different workflow: the agent must discover relevant documents, inspect metadata and file structure, choose the correct files, analyze structured and unstructured sources, and often move between **text and tables** within the same reasoning chain.
>
> For example, one simple task in lakeQA is:
>
> ```json
> "1": {
>   "source": "datagov/vsrr-provisional-drug-overdose-death-counts/files/rows.txt",
>   "fact": "In the VSRR provisional drug overdose death counts dataset, filtering to Indicator = 'Number of Drug Overdose Deaths', Period = '12 month-ending', Month = 'December', excluding State = 'US', and summing Data Value by year for 2015-2023 shows the highest total is in 2022.",
>   "subquestion": "For each year from 2016 through 2023, compute a growth-amplified overdose burden score: annual drug overdose deaths across all U.S. states multiplied by (1 + year-over-year growth rate).",
>   "answer": "2021"
> }
> ```
>
> This example illustrates the difference above. First, to obtain the answer, a complex query involving filter, aggregation and ranking is required. Second, the answer cannot be obtained by simply extracting a fact: the agent must determine which statistic is being asked for, how it should be aggregated, based on the auxilary documents, i.e. the metadata. Particularly, it must filter by **Month = December**, because that column represents a rolling 12-month sum; using other months would double count the quantity. We will make this contrast with BrowseComp more explicit and add more concrete cross-modal examples in the appendix.
>
> **2. More detailed experimental analysis.**
> To address the request for more granular analysis, we added a trajectory-level failure breakdown and tool-use statistics. We categorize failures into **Search Dataset Missing**, **Retrieved but Not Selected**, **Retrieved but Never Analyzed**, and **Analyzed but Still Wrong**:
>
> | Failure Type | GPT-5.2 | GPT-5-mini | Claude Sonnet 4.5 |
> |---|---:|---:|---:|
> | Correct | 19 | 10 | 26 |
> | Search Dataset Missing | 41 | 46 | 37 |
> | Retrieved but Not Selected | 12 | 8 | 11 |
> | Retrieved but Never Analyzed | 7 | 1 | 2 |
> | Analyzed but Still Wrong | 56 | 70 | 59 |
>
>   Here, **Search Dataset Missing** means there exists gold document that does not appear in any search results; **Retrieved but Not Selected** means there exists a gold document in search results but was not chosen for more detailed analysis; **Retrieved but Never Analyzed** means relevant data was inspected/downloaded but not queried; and **Analyzed but Still Wrong** means the agent queried all golden documents but still produced the wrong answer. This shows that many failures occur before successful document-level analysis begins. In the final version, we will add a clearer discussion of why each tool exists and how tool-format errors differ from retrieval-intent failures.
>
> We also added an evaluation of an official **deep research agent**. We evaluated **o4-mini-deep-research** on LAKEQA-mini as an official tool-enabled system rather than through our fixed tool interface. We provided each question directly and exposed our search tool through an MCP server, extracted the final submitted answer.
>
>   | Setting | Result |
>   |---|---:|
>   | Total tasks | 135 |
>   | Did not complete the task within 15 min | 109 |
>   | Completed tasks | 26 |
>   | Correct among completed | 9 |
>   | Wrong among completed | 17 |
>
> Among the 17 wrong completed cases, all had **source recall = 0.0** and **source precision = 0.0**, and none executed a document from the data lake. This further supports our conclusion that the main bottleneck is data discovery rather than downstream reasoning over retrieved evidence. We will also include additional analyses in the final version to provide a more comprehensive evaluation.
>
> **3. Typographical issue.**
> Thank you for catching the incomplete sentence before Section 3.4. We will fix it in the revision.

---

> > ### Author Rebuttal · Reviewer_sVyN · 2026-04-01
> >
> > Thank you for your response. I have increased the score to a 4. Since the authors emphasize that LakeQA is more analytic than extractive, I suggest including a comparison with other open-web benchmarks that share a similar analytical focus, such as FinSearchComp [1].
> >
> > [1] FinSearchComp: Towards a Realistic, Expert-Level Evaluation of Financial Search and Reasoning, ICLR 2026

---

### Official Review · Reviewer_wagW · 2026-03-03

**Soundness:** 2
**Presentation:** 4
**Significance:** 4
**Originality:** 3
**Overall Recommendation:** 4
**Confidence:** 3

**Summary:**

**One-line Summary**.
This paper introduces LakeQA, a novel *large-scale, search-dense, multi-hop* benchmark for the question-answering (QA) task of LLMs.

**Limitation of Prior Work**.
The authors identify three limitations of existing QA benchmarks that evaluate search & reasoning capabilities of LLMs:
- Low search density: only requiring search over a small document corpus
- Limited data diversity: often focusing on a homogenous data type (e.g., text) rather than integrating heterogeneous types (e.g., table+text)
- Low reasoning complexity: only requiring reasoning over <=4 documents to answer questions

**Core Idea for Improving Search density, Data diversity, and Reasoning Complexity**.
Extensive human annotation is employed, especially for multi-hop question generation and verification of benchmark correctness.

**Effect**.
LakeQA substantially raises the bar for QA benchmarks through:
- Higher search density: requires searching over >1M corpus
- Enhanced data diversity: address heterogeneous data type (e.g., table+text)
- Advanced reasoning complexity: requires average. 7.67 documents to answer questions

**Compliance With Llm Reviewing Policy:**

Affirmed.

**Final Justification:**

Weak accept. The rebuttal has sufficiently addressed my concerns.

**Key Questions For Authors:**

In Figure 3, I find that the diverse sectors of the questions are highly interesting. Could the authors provide concrete example questions for each sector? Specifically, how are nodes interconnected across heterogeneous data (table-to-text) to facilitate multi-hop reasoning in these domains?

**Limitations:**

I noticed that the manuscript lacks a dedicated discussion on the study's limitations. I strongly encourage the authors to include such a section in the final version, particularly addressing benchmark maintenance and the scalability of manual human annotations. Including these points would provide a more balanced perspective and help guide future research toward sustainable benchmark development.

**Strengths And Weaknesses:**

**Strengths**.

*1. Strategic Positioning and Motivation*

The paper is excellently positioned by addressing the critical bottlenecks of existing QA benchmarks, such as low search density and limited reasoning complexity.

&nbsp;

*2. Rigorous Challenge for State-of-the-Art LLMs*

The experimental results -- which includes the most advanced models like GPT-5.2 and Claude 4.5 -- clearly demonstrate the significant performance gaps that remain, highlighting the benchmark's potential to guide future research in retrieval and multi-hop reasoning.

&nbsp;

*3. High-Quality Presentation and Visualization*

The manuscript is well-structured and highly readable. Specifically, Figure 2 enables a rapid and intuitive understanding of the proposed methodology.


&nbsp;

**Weaknesses**.

*1. Limited Evaluation and Metrics for RAG Capabilities*

While LakeQA offers significant potential for debugging LLM behaviors, the evaluations presented in the paper are somewhat narrow.

- *Lack of Advanced RAG Baselines:* The experiments do not include modern, sophisticated RAG architectures, such as agentic RAG or RAG with iterative feedback loops, which are standard for such complex multi-hop tasks.

- *Over-reliance on Exact Match (EM)*: As shown in Figure 1, answers like "PS 056 Lewis H. Latimer" are highly sensitive to the Exact Match (EM) metric. Given the complexity of the tasks, incorporating more robust evaluation frameworks—such as LLM-as-a-judge—would significantly enhance the reliability and credibility of the results beyond a single EM score.

&nbsp;

*2. Insufficient Error Analysis for Tool-use in RQ2*

The authors claim in RQ2 that the primary bottleneck for LLM failure lies in the retrieval step. However, this claim is not sufficiently supported by a granular breakdown of the "search(query)" tool-use performance.

For example, it is unclear whether retrieval failures stem from (1) *Fundamental Reasoning Failures*, where the model fails to understand the multi-hop logic required to find the gold data, or (2) *Surface-level Syntactic Errors*, where the model's reasoning is correct, but it fails to execute the tool due to minor formatting issues (e.g., returning the wrong data type required in tool calls).

Without analyzing whether retrieval failures stem from logical errors or simple syntactic mistakes in tool-calling, the conclusion that "LLMs are unable to find gold datasets" remains somewhat speculative.

&nbsp;

*3. Static Nature and Maintenance Challenges*

As world knowledge evolves, a static benchmark constructed with high manual labor risks becoming outdated. The paper would benefit from providing guidelines for periodic updates or a discussion on how to mitigate the risk of incorrectness from oudated knowledge, ensuring the benchmark's long-term utility.

&nbsp;

---

> ### Author Rebuttal · Authors · 2026-03-31
>
> We thank the reviewer for the positive assessment of the paper’s positioning, challenge level, and presentation. We address each point below.
>
> **1. RAG baselines and evaluation metrics.**
> We added **RAG-style retrieval baselines** over **10K, 25K, and 50K** document subsets, under **BM25** and **Hybrid search**:
>
> | Model | 10K | 25K | 50K |
> |---|---:|---:|---:|
> | Claude Sonnet 4.5, BM25 | 37.04 | 31.11 | 35.56 |
> | Claude Sonnet 4.5, Hybrid | 32.59 | 34.07 | 33.33 |
> | Haiku 4.5, BM25 | 19.26 | 17.78 | 19.26 |
> | Haiku 4.5, Hybrid | 24.44 | 15.56 | 14.81 |
>
> All values are EM (%). Stronger retrieval helps in some settings but does not remove the core challenge. RAG in our setting is also materially different from standard text-only RAG: embedding a 9.5 TB heterogeneous data lake is expensive, and tables introduce additional design choices about what to embed, such as metadata, schema, rows, or chunked content. We therefore view the current BM25 and hybrid results as initial baselines rather than an exhaustive exploration of the RAG design space.
>
> We also analyzed answer types in LAKEQA-mini: **62/135 (45.9%)** answers are **numeric** and **33/135 (24.4%)** are **temporal**, so together these two categories account for **95/135 (70.4%)** of all tasks. Many questions also specify explicit formats such as “round to the nearest integer” or “report in MM/DD format,” which is why case-insensitive exact match is a reasonable primary metric. We also evaluated **LLM-as-a-judge** and **human evaluation**:
>
> | Model | Exact Match | LLM-as-a-Judge | Human |
> |---|---:|---:|---:|
> | Claude Sonnet 4.5 | 26/135 | 27/135 | 27/135 |
> | OpenAI GPT-5.2 | 19/135 | 21/135 | 21/135 |
> | OpenAI GPT-5-mini | 10/135 | 10/135 | 10/135 |
>
> Across all three models, EM failed only in three surface-variation cases: “the Cut Bank Penguin” vs. “Cut Bank Penguin,” “Pittsburgh, Pennsylvania” vs. “Pittsburgh, PA,” and “Accomack County” vs. “Accomack.” We plan to add LLM-as-a-judge as a supplementary metric alongside exact match in the camera-ready version if accepted.
>
> **2. More granular error analysis for RQ2.**
> We added a trajectory-level breakdown with four categories: **Search Dataset Missing**, **Retrieved but Not Selected**, **Retrieved but Never Analyzed**, and **Analyzed but Still Wrong**.
>
> | Failure Type | GPT-5.2 | GPT-5-mini | Claude Sonnet 4.5 |
> |---|---:|---:|---:|
> | Correct | 19 | 10 | 26 |
> | Search Dataset Missing | 41 | 46 | 37 |
> | Retrieved but Not Selected | 12 | 8 | 11 |
> | Retrieved but Never Analyzed | 7 | 1 | 2 |
> | Analyzed but Still Wrong | 56 | 70 | 59 |
>
> Here, **Search Dataset Missing** means there exists gold document that does not appear in any search results; **Retrieved but Not Selected** means there exists a gold document in search results but was not chosen for more detailed analysis; **Retrieved but Never Analyzed** means relevant data was inspected/downloaded but not queried; and **Analyzed but Still Wrong** means the agent queried all golden documents but still produced the wrong answer. This shows that many failures occur before successful document-level analysis begins. We will expand this into a fuller per-tool breakdown in the final version.
>
> **3. Concrete examples and static maintenance challenges.**
> We will add representative example questions for each sector in the appendix. One analytic example is:
>
> ```json
> "1": {
>   "source": "datagov/vsrr-provisional-drug-overdose-death-counts/files/rows.txt",
>   "fact": "In the VSRR provisional drug overdose death counts dataset, filtering to Indicator = 'Number of Drug Overdose Deaths', Period = '12 month-ending', Month = 'December', excluding State = 'US', and summing Data Value by year for 2015-2023 shows the highest total is in 2022.",
>   "subquestion": "For each year from 2016 through 2023, compute a growth-amplified overdose burden score: annual drug overdose deaths across all U.S. states multiplied by (1 + year-over-year growth rate).",
>   "answer": "2021"
> }
> ```
>
> Solving this is not a simple fact lookup: the agent must identify the correct statistic, apply the right filters, and interpret metadata correctly. In particular, it must filter by **Month = December**, because that column corresponds to a rolling 12-month sum; failing to do so leads to double counting.
>
> We will also add a dedicated limitations section. LAKEQA is built on fixed snapshots of Wikipedia and Data.gov so that evaluation is reproducible, but this creates a tradeoff: as the sources evolve, a fixed snapshot becomes less current. Our plan is to keep the current release as a stable, versioned benchmark and handle future changes through separately versioned refreshes with documented source snapshots and task sets. This preserves reproducibility while providing a clear mechanism for periodic updates. Similarly, manual annotation ensured task quality and multi-hop correctness in the first release, but it limits scalability; we will make both tradeoffs explicit in the final version.

---

> > ### Author Rebuttal · Reviewer_wagW · 2026-04-04
> >
> > While there are some remaining concerns, I thank the authors for the detailed response. I will keep my positive score.

---

### Official Review · Reviewer_XeNH · 2026-03-12

**Soundness:** 3
**Presentation:** 3
**Significance:** 3
**Originality:** 3
**Overall Recommendation:** 4
**Confidence:** 3

**Summary:**

This paper introduces LAKEQA, a benchmark for exploratory question answering over a heterogeneous data lake of approximately 9.5 TB and 40 million documents from Wikipedia and Data.gov, where each task requires an agent to iteratively search for and reason over multiple documents (averaging 7.67 per task) to answer complex multi-hop questions. Experiments on seven frontier LLMs show that even the best model achieves only 23.08% exact match accuracy, with analysis revealing that the primary bottleneck is document discovery rather than reasoning over retrieved evidence.

**Compliance With Llm Reviewing Policy:**

Affirmed.

**Final Justification:**

LAKEQA addresses a genuine gap in QA benchmarking by combining high search density and reasoning density at a realistic data lake scale (40M documents, 9.5TB). The diagnostic framework separating search failure from reasoning failure is well-designed and yields actionable insights.

The rebuttal thoroughly addressed my concerns. The detailed annotation process description (9 annotators, multi-stage construction/rewrite/verification pipeline) clarifies the quality control measures. The o4-mini-deep-research experiment — where 109/135 tasks timed out and all 17 wrong completions had zero source recall — reinforces the finding that document discovery is the primary bottleneck. The EM vs. LLM-judge vs. human evaluation comparison (only 3 surface-variation discrepancies across 3 models) convincingly validates exact match as a reasonable metric, and the fixed-snapshot reproducibility plan is sensible.

I maintain my score of 4 (Weak Accept).

**Key Questions For Authors:**

1. How many annotators were involved in total, and what is the inter-annotator agreement rate? A low agreement rate would raise concerns about benchmark reliability.

2. Have you considered evaluating dedicated deep research agents (e.g., OpenAI Deep Research and Tongyi Deep Research) on LAKEQA? If so, were there practical barriers, and do you plan to include such baselines?

3. What is the precise answer format expected by the exact match metric? Are there cases where semantically correct answers are penalized due to surface-level mismatches, and if so, how prevalent is this issue?

4. The data lake relies on a specific Wikipedia dump and a Data.gov archive. How do you plan to handle temporal drift as these sources are updated, and is the benchmark designed to be reproducible with fixed snapshots?

**Limitations:**

yes

**Strengths And Weaknesses:**

**Strengths:**

1. The benchmark addresses a genuine gap by combining high search density and high reasoning density at a realistic data lake scale of 40 million documents, which is significantly larger and more heterogeneous than existing QA benchmarks. This makes LAKEQA well-suited for evaluating the kind of iterative search-and-reasoning workflows that arise in real-world enterprise data analysis.

2. The experimental evaluation is thorough, covering seven frontier LLMs with a well-designed diagnostic framework that separates search failure from reasoning failure through the retrieval set and accessed set metrics. The analysis that document discovery is the primary bottleneck, rather than reasoning over found evidence, provides actionable insight for future agent development.

**Weaknesses:**

1. The paper provides limited information about annotators. Key details such as the total number of annotators, their recruitment process, compensation, and inter-annotator agreement metrics are not reported. While the paper mentions that each task is reviewed by at least three annotators including one CS Ph.D. student, the lack of these standard annotation transparency measures makes it difficult to fully assess the reliability and reproducibility of the benchmark.

2. The evaluation does not include any dedicated deep research agents, whether open-source or commercial. Systems such as OpenAI's Deep Research and similar products are specifically designed for iterative search and long-horizon reasoning, making them natural baselines for this benchmark, and their absence leaves an important gap in understanding how well current purpose-built exploration agents perform on LAKEQA.

3. The paper does not explicitly define the expected answer format, such as whether answers are short entities, numerical values, or free-form text. This is somewhat inconsistent with the use of exact match as the sole end-to-end evaluation metric, since EM is sensitive to surface-level variations and its appropriateness depends heavily on the answer type. Clarifying the answer format and discussing potential limitations of EM, or supplementing it with softer metrics, would strengthen the evaluation.

---

> ### Author Rebuttal · Authors · 2026-03-31
>
> We thank the reviewer for the constructive suggestions. We address each point below.
>
>   **1. Annotation process and reliability.**
>   To address this concern, we provide a more detailed description of the annotation process here and will make it clearer in the final version of the paper. LAKEQA uses a
>   two-stage pipeline: **question construction** and **question rewrite**. Our annotation team consisted of **nine people**: **five CS PhD students** with data management
>   research experience, and **four senior CS undergraduates** selected from a pool of about **30 students** based on a manual task-construction exercise and interview
>   performance. Undergraduate annotators were compensated under standard university undergraduate worker policies.
>
>   In construction, a PhD annotator selects a source document, extracts a fact, turns it into a subquestion-answer pair, and extends the reasoning chain. In rewrite, an
>   undergraduate annotator rewrites the composed chain into a single natural-language question while preserving meaning and avoiding information leakage. Each task is
>   checked by two undergraduates and one PhD annotator, and each task is reviewed by at least **four annotators**, including at least **two PhD annotators**; tasks that fail
>   a check are revised or removed. Because this is a benchmark construction pipeline rather than a standard flat labeling task, conventional inter-annotator agreement is not
>   the main quality-control measure here. We instead rely on multi-stage construction, rewriting, verification, and expert review.
>
>   **2. Deep research agents.**
>   We added a new experiment on **OpenAI o4-mini-deep-research**, evaluated as an official tool-enabled system rather than through our fixed benchmark interface. We provided
>   each LAKEQA-mini question directly to the system, exposed benchmark search through an MCP server, extracted the final submitted answer, and evaluated it against the
>   LAKEQA-mini ground truth using the same metrics as in the rest of the paper.
>
>   | Setting | Result |
>   |---|---:|
>   | Total tasks | 135 |
>   | Did not complete the task within 15 min | 109 |
>   | Completed tasks | 26 |
>   | Correct among completed | 9 |
>   | Wrong among completed | 17 |
>
>   Among the 17 wrong completed cases, all had **source recall = 0.0** and **source precision = 0.0**, and none executed a document from the data lake. In **9** cases, the agent
>   never reached any benchmark source; in the other **8**, it reached related but non-gold sources. This suggests that the main failure mode was source discovery and source
>   selection rather than downstream reasoning over correctly retrieved evidence.
>
>   **3. Answer format and evaluation metric.**
>   We will clarify the answer format in the revision. We additionally analyzed the answer-type distribution in LAKEQA-mini. Most answers are constrained outputs: **62/135
>   (45.9%)** are **numeric** and **33/135 (24.4%)** are **temporal**, so together these two categories account for **95/135 (70.4%)** of all tasks. For many of these
>   questions, we also specify explicit answer formats such as “round to the nearest integer” or “report in MM/DD format.” This is why case-insensitive exact match is a
>   reasonable primary metric for LAKEQA-mini.
>
>   To understand the impact of non-EM metrics, we also evaluated LAKEQA-mini using **(1) case-insensitive exact match**, **(2) LLM-as-a-judge**, and **(3) human
>   evaluation**:
>
>   | Model | Exact Match | LLM-as-a-Judge | Human |
>   |---|---:|---:|---:|
>   | Claude Sonnet 4.5 | 26/135 | 27/135 | 27/135 |
>   | OpenAI GPT-5.2 | 19/135 | 21/135 | 21/135 |
>   | OpenAI GPT-5-mini | 10/135 | 10/135 | 10/135 |
>
>   Across all three models, EM failed only in three surface-variation cases: “the Cut Bank Penguin” vs. “Cut Bank Penguin”, “Pittsburgh, Pennsylvania” vs. “Pittsburgh, PA”,
>   and “Accomack County” vs. “Accomack”. We plan to add LLM-as-a-judge as an additional metric alongside exact match in the camera-ready version if accepted.
>
>   **4. Temporal drift and reproducibility.**
>   Yes. LAKEQA is built on fixed snapshots: a specific **English Wikipedia dump from December 20, 2025** and a specific **Harvard LIL Data.gov archive snapshot**, with
>   stable dataset identifiers. This means all models are evaluated against the same evidence pool, file structure, and source state, making comparisons reproducible.
>
>   Temporal drift is an important limitation as these sources evolve. Our plan is to keep the current release fixed as a stable, versioned benchmark and handle future
>   changes through separately versioned refreshes with documented source snapshots and task sets. We will make this tradeoff explicit in the paper.

---

> > ### Author Rebuttal · Reviewer_XeNH · 2026-03-31
> >
> > I thank the authors for their detailed rebuttal. The responses adequately address most of my concerns. I maintain my overall recommendation.

---

### Official Review · Reviewer_fMmK · 2026-03-12

**Soundness:** 3
**Presentation:** 2
**Significance:** 2
**Originality:** 3
**Overall Recommendation:** 4
**Confidence:** 3

**Summary:**

This paper introduces LaKeQA, a large-scale, comprehensive exploratory question-answering benchmark dataset designed to evaluate the joint search and reasoning capabilities of LLM agents on a million-scale heterogeneous data lake, along with providing an agentic tool suite for agents.

**Compliance With Llm Reviewing Policy:**

Affirmed.

**Final Justification:**

The response is overall satifying, and I have increased my rating to 4.

**Key Questions For Authors:**

If the author has a clear open-source plan for the LAKEQA benchmark dataset?

Are there any rule-based relaxations applied to the EM calculation and are there other metrics tested?

**Limitations:**

Yes

**Strengths And Weaknesses:**

**Strengthes:**

Soundness:

This benchmark dataset follows clear design principles for heterogeneity, inference, searchability, and systematic task creation, including sub problem construction, chain connection, rewriting, and other processes.

Presentation:

This paper provides an excellent illustration of the EQA process, provides a detailed explanation of the dataset construction process, and supplements annotation guidelines.

Significance:

This paper deepens our understanding of the locality of LLM agents in joint search and inference, and empirically finds that document discovery is the main bottleneck.

The QA scenarios in the real world rarely provide predefined evidence, and useful information is often scattered in massive and heterogeneous data lakes. This benchmark dataset fills the gap in existing EQA evaluations.

Originality:

This paper aims to evaluate LLM agents on EQA over a massive, million-scale data lake. While existing benchmarks focus on multi-hop reasoning over curated, small-scale contexts, LAKEQA forces agents to navigate a 9.5 TB heterogeneous environment

**Weaknesses:**

1. There is no open-source benchmark dataset and accompanying experimental code available.

2. The paper points out that performance decreases with increasing inference density, but lacks fine-grained ablation analysis on where agents usually fail first.

3. Further, as a benchmark paper, the experimental evaluation currently feels somewhat limited. Beyond reporting overall results, it would be helpful to include more comprehensive analyses of dataset coverage, difficulty breakdown, error types, and robustness across different settings. Stronger and more representative baselines (if there are some data agents designed for this task) would also help better establish the value of the benchmark.

---

> ### Author Rebuttal · Authors · 2026-03-31
>
> We thank the reviewer for the positive assessment and constructive suggestions. We address each point below.
>
>   **1. Open-source plan.**
>   We plan to fully open-source LAKEQA. The release will include the benchmark tasks, the chains describing how each task can be solved, the evaluation suite, and a live
>   leaderboard. For review purposes, we provide anonymous access to LAKEQA-mini and the current evaluation suite to reviewers, and will publicly release the full benchmark
>   package after the review process. We provide the anonymous repo at https://anonymous.4open.science/r/LakeQA-ICML-D0D5 containing the LakeQA-mini dataset we used in the rebuttal.
>
>   **2. Fine-grained analysis of experimental results.**
>   To address this point, we added a trajectory-level breakdown on LAKEQA-mini with four failure modes. Below we show results for GPT-5.2, GPT-5-mini, and Claude Sonnet 4.5;
>   we will include the full analysis for all evaluated models in the final version.
>
>   | Failure Type | GPT-5.2 | GPT-5-mini | Claude Sonnet 4.5 |
>   |---|---:|---:|---:|
>   | Correct | 19 | 10 | 26 |
>   | Search Dataset Missing | 41 | 46 | 37 |
>   | Retrieved but Not Selected | 12 | 8 | 11 |
>   | Retrieved but Never Analyzed | 7 | 1 | 2 |
>   | Analyzed but Still Wrong | 56 | 70 | 59 |
>
>   Here, **Search Dataset Missing** means there exists gold document that does not appear in any search results; **Retrieved but Not Selected** means there exists a gold document in search results but was not chosen for more detailed analysis; **Retrieved but Never Analyzed** means relevant data was inspected/downloaded but not queried; and **Analyzed but Still Wrong** means the agent queried all golden documents but still produced the wrong answer. This shows that many failures occur before successful document-level analysis begins.
>
>   **3. Broader evaluation and stronger baselines.**
>   We added new experiments under different retrieval and evaluation settings.
>
>   First, we added **RAG-style retrieval baselines** over **10K, 25K, and 50K** document subsets, under **BM25** and **Hybrid search** (keyword + vector retrieval):
>
>   | Model | 10K | 25K | 50K |
>   |---|---:|---:|---:|
>   | Claude Sonnet 4.5, BM25 | 37.04 | 31.11 | 35.56 |
>   | Claude Sonnet 4.5, Hybrid | 32.59 | 34.07 | 33.33 |
>   | Haiku 4.5, BM25 | 19.26 | 17.78 | 19.26 |
>   | Haiku 4.5, Hybrid | 24.44 | 15.56 | 14.81 |
>
>   All values are EM (%). These results show that stronger retrieval support helps in some settings but does not remove the core challenge.
>
>   Second, we added an evaluation of an official **deep research agent**. We evaluated **o4-mini-deep-research** on LAKEQA-mini as an official tool-enabled system rather
>   than through our fixed tool interface. We provided each question directly, exposed benchmark search through an MCP server, extracted the final submitted answer, and
>   evaluated it with the same metrics as the rest of the paper.
>
>   | Setting | Result |
>   |---|---:|
>   | Total tasks | 135 |
>   | Did not complete the task within 15 min | 109 |
>   | Completed tasks | 26 |
>   | Correct among completed | 9 |
>   | Wrong among completed | 17 |
>
>   Among the 17 wrong completed cases, all had **source recall = 0.0** and **source precision = 0.0**, and none executed a benchmark dataset file. In **9** cases, the agent
>   never reached any benchmark source; in **8**, it reached related but non-gold sources. This further supports our conclusion that the main bottleneck is source discovery
>   and source selection rather than downstream reasoning over retrieved evidence.
>
>   We will also include additional analyses in the final version to provide a more comprehensive evaluation.
>
>   **4. Exact match, relaxations, and alternative metrics.**
>   Our current end-to-end metric is **case-insensitive exact match**, and we do **not** apply task-specific rule-based relaxations. We also analyzed the answer-type
>   distribution in LAKEQA-mini: **62/135 (45.9%)** answers are **numeric** and **33/135 (24.4%)** are **temporal**, so together these categories account for **95/135
>   (70.4%)** of all tasks. Many of these questions also specify explicit answer formats such as “round to the nearest integer” or “report in MM/DD format,” which is why
>   exact match is a reasonable primary metric.
>
>   To understand the impact of non-EM metrics, we also evaluated LAKEQA-mini using **(1) case-insensitive exact match, (2) LLM-as-a-judge, and (3) human evaluation**:
>
>   | Model | Exact Match | LLM-as-a-Judge | Human |
>   |---|---:|---:|---:|
>   | Claude Sonnet 4.5 | 26/135 | 27/135 | 27/135 |
>   | OpenAI GPT-5.2 | 19/135 | 21/135 | 21/135 |
>   | OpenAI GPT-5-mini | 10/135 | 10/135 | 10/135 |
>
>   Across all three models, EM failed only in three surface-variation cases: “the Cut Bank Penguin” vs. “Cut Bank Penguin”, “Pittsburgh, Pennsylvania” vs. “Pittsburgh, PA”,
>   and “Accomack County” vs. “Accomack”. We plan to add LLM-as-a-judge as an additional metric alongside exact match in the camera-ready version if accepted.

---

> > ### Author Rebuttal · Reviewer_fMmK · 2026-04-01
> >
> > Thanks for the response. I will increase my rating to 4.

---

### Decision · Program_Chairs · 2026-04-30

**Decision:**

Accept (regular)

**Comment:**

LakeQA is a large-scale exploratory question answering (EQA) benchmark spanning a 9.5 TB data lake of Wikipedia and Data.gov documents. It requires agents to iteratively search and reason over an average of 7.67 documents to solve complex multi-hop questions. Expert-annotated via a multi-stage pipeline, the benchmark is exceptionally challenging. Diagnostic analysis identifies document discovery, rather than reasoning, as the primary performance bottleneck.

Principal Merits of the Benchmark:
It addresses a significant research lacuna by integrating intensive search and reasoning requirements.
The framework successfully distinguishes between search-related and reasoning-based deficiencies through a rigorous diagnostic methodology.
It establishes a high-fidelity ground truth, substantiated by expert annotations at the doctoral level.
It delivers substantive observations regarding the constraints of current document discovery processes.

This benchmark represents a significant and timely advancement in the field of Large Language Model (LLM) evaluation, promising enduring relevance through its substantial level of difficulty. A comprehensive rebuttal effectively resolved all reviewer inquiries by incorporating trajectory-level analysis and examining advanced research agents. Furthermore, the dedication to facilitating community access through a public leaderboard and versioned snapshots underscores its utility to the ICML scholarly community.